



# A 17-year dataset of surface water fugacity of CO₂, along with calculated pH, Aragonite saturation state, and air-sea CO₂ fluxes in the Northern Caribbean Sea

Rik Wanninkhof[1], Denis Pierrot[1,2], Kevin Sullivan[1,2], Leticia Barbero[1,2], and Joaquin Triñanes[1,2,3]

[1]Atlantic Oceanographic and Meteorological Laboratory (AOML), NOAA, 4301 Rickenbacker Causeway, Miami, FL 33149 USA

[2]Cooperative Institute for Marine and Atmospheric Studies, Rosenstiel School of Marine and Atmospheric Science, University
of Miami, 4600 Rickenbacker Causeway, Miami, FL 33149 USA

[3]Laboratory of Systems, Technological Research Institute, Universidad de Santiago de Compostela, Campus Universitario Sur, Santiago de Compostela, 15782, Spain

*Correspondence to*: Rik Wanninkhof (rik.wanninkhof@noaa.gov)





**Abstract.** A high-quality dataset of surface water partial pressure/fugacity of $CO_2$ ($pCO_{2w}/fCO_{2w}$)[1], comprised of over a million

observations, and derived products are presented for the Northern Sea covering the timespan from 2002 through 2018. Prior to installation of automated $pCO_2$ systems on cruise ships of the Royal Caribbean Cruise Lines and subsidiaries, very limited surface water carbon data were available in this region. With this observational program, the Northern Caribbean Sea has now become one of the best sampled regions for $pCO_2$ of the world's ocean. The dataset, and derived quantities are provided on a 1-degree monthly grid at http://accession.nodc.noaa.gov/0207749, DOI:10.25921/2swk-9w56 (Wanninkhof et al., 2019a). The derived quantities include total alkalinity (TA), acidity (pH), Aragonite saturation state ($\Omega_{Ar}$) and air-sea $CO_2$ flux, and cover

the region from 15° N to 28° N and 88° W to 62° W. The data and products are used for determination of status and trends of ocean acidification, for quantifying air-sea $CO_2$ fluxes, and for ground truthing models. Methodologies to derive the inorganic carbon system parameters and to calculate the fluxes from $fCO_{2w}$ are described.

**Introduction**

Over the past 20 years a rapidly expanding program of measurements of surface water partial pressure of carbon dioxide

($pCO_{2w}$), or the fugacity of $CO_2$ ($fCO_{2w}$), has provided data to determine air-sea $CO_2$ fluxes, and rates of ocean acidification on local to global scales (e.g. Boutin et al., 2008; Degrandpre et al., 2002; Evans et al. 2015; Schuster et al., 2012; Takahashi et al., 2014; Wanninkhof et al. 2019 a, b). Marginal seas, that historically had a dearth of measurements, have been targeted for increased observations. Through an industry, academic, and federal partnership between the U.S. National Oceanic and Atmospheric Administration (NOAA), Royal Caribbean Cruise Lines (RCCL), and the University of Miami, cruise ships were

outfitted with automated surface water $pCO_2$ systems, also called underway $pCO_2$ systems (Pierrot et al., 2009). In 2002, the RCCL ship *Explorer of the Seas* (*EoS*) was equipped with an underway $pCO_2$ setup providing observations on alternating weekly transects from Miami, FL to the (North) Eastern and Western Caribbean. In 2008, the home port for *EoS* was shifted to Bayonne, New Jersey, and itineraries were shared between the Eastern Caribbean, Bermuda, and Eastern US. In 2015, the *EoS* was repositioned out of the Atlantic and the *Celebrity Equinox* (*Eqnx*) was outfitted with an underway $pCO_2$ system.

Additionally, the *Allure of the Seas (AoS)* was outfitted with an underway $pCO_2$ system in 2016. The *Eqnx* and *AoS* covered similar transects as the *EoS* but on more irregular and seasonal basis. A total of 582 cruises covered the region from 2002 through 2018. A map of all cruise tracks is shown in **Figure 1**. The number of cruises per year covering the Caribbean Sea and adjacent Western Atlantic are provided in **Figure 2**. The bar graph shows fewer cruises in the middle part of the record when the *EoS* was diverted to other routes outside the area, and eventually repositioned.

---

[1] $fCO_2$ is the $pCO_2$ corrected for the non-ideal behavior of $CO_2$ (Weiss 1974). In surface water $fCO_{2w} \approx 0.997\ pCO_{2w}$



The surface water pCO₂ observational dataset and derived products including total alkalinity (TA), acidity (pH), aragonite saturation state ($\Omega_{Ar}$), and air-sea $CO_2$ flux are of importance for determining the anthropogenic carbon uptake, and to assess trends and impacts of ocean acidification. The observational data are provided to the global surface ocean carbon atlas (SOCAT) (Bakker et al., 2016) and global $CO_2$ climatology (Takahashi et al., 2009; 2018). These data are the main source of $fCO_2$ observations available in the region, and the high frequency of measurements provides a seasonally resolved picture of

changing surface water $fCO_2$ ($fCO_{2w}$). This effort has made the Northern Caribbean one of the few places in the world's ocean where such regional observational density has been established. The data and mapped products are used in Wanninkhof et al. (2019b) who show large decadal changes in trends of surface water $fCO_2$ and associated changes in air-sea $CO_2$ fluxes.

The data from the first part of this record form the basis of the Caribbean ocean acidification product suite that maps ocean acidification conditions in the Caribbean (Gledhill et al., 2008; https://www.coral.noaa.gov/accrete/oaps.html;

https://cwcgom.aoml.noaa.gov/erddap/griddap/miamiacidification.graph). The large, high quality, and well-resolved dataset is also used to validate models (Gomez et al., 2018).

For optimal application, datasets and associated data products are fully documented here, and are readily accessible according to the findable, accessible, interoperable and reusable (FAIR) principles (Wilkinson et al., 2016). The documentation of the cruises, the sampling methodology, and data reduction techniques are presented. This is followed by a description of the

approaches to calculate the different inorganic carbon system parameters. The procedure is to estimate the so-called second inorganic carbon system parameter that is used in conjunction with $fCO_{2w}$. In this case we estimate the total alkalinity (TA) based on robust relationships of TA with salinity (Cai et al., 2010; Takahashi et al., 2014; Lee et al., 2006; Millero et al., 1998), and use a software program CO2SYS (Pierrot et al., 2006) to calculate the other inorganic carbon system parameters of interest, in this case pH and $\Omega_{Ar}$. These data products are presented at monthly scales binned and averaged on a 1˚ by 1˚ grid. Annual

multi-linear regressions (MLR) are developed between the binned $fCO_{2w}$ data, and sea surface temperature (SST), sea surface salinity (SSS), location (latitude and longitude), and mixed layer depth as independent variables. These regressions are applied at 1˚ by 1˚ spatial resolution to the region of interest between 15˚ N-28˚ N and 88˚ W-62˚ W using remotely sensed or modeled independent variables. The procedures and datasets, including tables of column headers of the product created, are detailed below.

**1 Observations**

The observational program is described in terms of the ships, voyages, and instrumentation. The ships predominantly sailed in the Caribbean Sea, but also had tracks outside the region, including in the Northeast of the USA, to Bermuda and in the Mediterranean. The description and data presented here cover the Northern Caribbean Sea and the Western Atlantic (Fig. 1), collectively called the Caribbean. The homeports of the ships, where passengers embark and disembark, are Miami and Fort

Lauderdale, FL. The *Explorer of the Seas (EoS)* changed homeport from Miami, FL to Cape Liberty Cruise Port, NJ in 2008, and changed its routes at that point to include cruises with Bermuda as a port of call. In 2015 the *EoS* was repositioned to the



Pacific and the underway $pCO_2$ system was removed. From 2015 onward, the *Celebrity Equinox (Eqnx)* and from 2016 onward the *Allure of the Seas (AoS)* covered the area. The *Eqnx* spent the summers of 2015 and 2016 in the Mediterranean, causing data gaps in the Caribbean.

## 1.2 Cruises

The underway $pCO_2$ systems were installed on three different ships of the Royal Caribbean Cruise Lines. The *EoS* had 331 cruises from 2002 to 2015; the *Eqnx* completed 135 cruises from 2015 through 2018; and the *AoS* performed 116 cruises in the study area from 2016 through 2018. The coverage represents the Northern Caribbean Sea and Western Atlantic, north of the Caribbean islands chain. Temporal coverage over the 16 years shows at least bi-weekly occupations at the beginning of the record from 2002 to 2007, and at the end from 2014 through 2018, with fewer occupations in the years in between (Figure 2). The cruises lasted between 7 and 14 days and made about a half a dozen ports of call. The ships generally were in port from early morning to late afternoon, and transited between ports at night, except for long runs (e.g., from Miami to San Juan) when the ship sailed continuously for several days. Ports are listed in the metadata accompanying the original data.

Installation of the systems occurred in different locations for the three ships, but each had a dedicated seawater intake near the bow of the ship. The *EoS* had an intake in the bow thruster tube (≈3 m depth) that was non-optimal due to bubble entrainment during bow thruster operations and heavy seas. These observations have been culled from the datasets. The *AoS* and *Eqnx* also have their intake at ≈5 m depth, but forward of the bow thruster and have had fewer issues with bubble entrainment. On the *EoS,* the underway $pCO_2$ instrument was initially in a dedicated science laboratory built for purpose on the ship, and located amidships about 100m from the intake. In 2008, a new system was placed in the engineering space closer to the bow with no apparent change in performance. For the *AoS* and *Eqnx,* the instruments were near the bow intake in the engineering space, about 5-m from the intake. Seawater flow to the systems was about $4 \, l \, min^{-1}$. The *EoS* had an air intake mounted in August of 2008 on a mast at the forward most point of the main deck. The *Eqnx* had an air intake near the bow one level below the main deck since the initial installation in March 2015. The underway $pCO_2$ systems on these ships thus made marine boundary layer (MBL) air observations ($xCO_{2a}$) but these are not used in the data products presented.

## 1.3 Instrumentation

### 1.3.1 pCO₂ system

The instrumentation is based on a community design described in Pierrot et al. (2009). The instruments are manufactured by General Oceanics Inc. and have performed to high accuracy specifications (Wanninkhof, 2013). The surface water drawn from the intake goes through a 1.1-l sprayhead equilibrator with a water volume of 0.5-l and a headspace of 0.6-l. The spray and agitation causes the $CO_2$ in the headspace to equilibrate with the $CO_2$ in water with a response time of about 2 minutes (Pierrot et al., 2009; Web et al., 2016). Thus, the air in the headspace will reach 99.8 % equilibration in 12 minutes. As the air is recirculated and surface waters are relatively homogeneous on hourly timescales, equilibration can be assumed under most

circumstances. Typical cruise speeds are 22 knots which, with sampling every 2.5 minutes, yields a sample approximately every 1.7 km except for 35 minutes every 4.5 hours when four calibration gases and a $CO_2$-free reference gas are analyzed,

followed by 5 atmospheric $CO_2$ measurements for the ships with air intakes. This creates a gap of 24 km (≈1/4˚) without $fCO_{2w}$ measurements. The systems are automatically turned off when the ships enter port, and upon shutdown they are back-flushed with fresh water removing particles from the inline water filter thereby alleviating clogging issues in the filter and reducing biofouling of water lines, filter and equilibrator.

The $fCO_{2w}$ levels are measured by infrared (IR) analysis of the headspace gas in the equilibration chamber. The gas in the

headspace is partially dried (≈ >75%) before measurements. The instruments have shown agreement to within 1 µatm with other state-of-art systems in intercomparison studies (Nojiri et al., Feb. 2009 pers. com.). The IR sensor (LI-COR 6262) is calibrated every 4.5 hours against four standard gases supplied by the global monitoring division of the environmental science research laboratory of NOAA (GMD/ ESRL/NOAA), traceable to the WMO $CO_2$ mole fraction scale. The $CO_2$ concentrations of the standards span the range of surface water values encountered along the ship tracks. Several different versions of the

instrument have been deployed on the ships over the years but overall measurement principles and accuracies, estimated at better than 2 µatm, have been maintained.

### 1.3.2 Thermosalinograph

Temperature and salinity are measured with a flow-through Seabird SBE45 thermosalinograph that is in a seawater flow line

parallel to the $pCO_2$ equilibrator. A SBE38 remote temperature probe is situated near the inlet before the pump, and is used as the SST measurement. The thermosalinograph and temperature probes were maintained by collaborators from the Marine Technical group at the Rosenstiel School of Marine and Atmospheric Sciences at the University of Miami (RSMAS/U. Miami) and swapped out annually for calibration.

### 1.3.3 Other instrumentation


The underway effort is part of a larger scientific operation lead by RSMAS called Oceanscope (https://oceanscope.rsmas.miami.edu/). Additional instrumentation onboard the ships include the Marine-Atmospheric Emitted Radiance Interferometer (M-AERI) to assess surface skin temperature retrievals from a number of radiometers on earth-observation satellites (Minnet et al., 2001). The M-AERI's are Fourier-Transform Infrared interferometers situated on

the deck viewing the sea surface away from the wake. Acoustic Doppler Current Profilers (ADCP) mounted on the hull of the ships, are used to measure ocean currents.





## 2 Datasets used

### 2.1 pCO₂ data

Full details on data acquisition and calculation of $pCO_2$ and $fCO_2$ can be found in Pierrot et al. (2009). Briefly, the $pCO_2$ data was acquired at approximately 2.5 minute intervals in a continual sequence. The sequence included 4 reference standards spanning the range of surface water values ($\approx$ 280- 480 ppm), 5 marine boundary layer (MBL) air measurements (for the ships with an air inlet), and 100 equilibrator headspace samples, henceforth called water samples. The full sequence took about 4.5 hours. The gas entering the analyzer was dried by passing them through a thermo-electric cooler at 5˚ C and a PermaPure drier.

The standards were bone dry; both air and water analyses typically had about 10 % or less humidity. Every 27 hours the LICOR model 6262 infrared analyzer was zeroed with the $CO_2$-free air and spanned with the highest standard. The dry mole fraction of $CO_2$ ($xCO_2$ in parts per million, ppm), as calculated and output by the analyzer based on measured $CO_2$ and water vapor levels, were recorded. Two weeks of output data from the *AoS* and *Eqnx* are displayed online in graphical format at https://www.aoml.noaa.gov/ocd/ocdweb/allure/allure_realtime.html and

https://www.aoml.noaa.gov/ocd/ocdweb/equinox/equinox_realtime.html, respectively.

Post-cruise, the $xCO_2$ data were processed by first linearly interpolating each standard measured every ~4 hours to the time of a sample measurement and then recalculating the air and water $xCO_2$ values based on the linear regression of the interpolated standard values at the time of sample measurement. Infrequently, the analyzer output negative water vapor values due to the condition of desiccant chemicals, and thus yielded erroneous dried $xCO_2$ values. Separate processing routines were developed

to correct for these situations (available on request from D. Pierrot). The post-cruise corrected $xCO_2$ values were used for calculation of $pCO_2$ and $fCO_2$ as described in the calculation section.

### 2.2 Thermosalinograph, sea surface temperature and salinity data

The SST and salinity data, obtained from temperature and conductivity measured in the thermosalinograph, were appended to the $pCO_2$ data records in real-time and also logged via another shipboard computer at a more frequent interval. The

thermosalinographs on the ships were factory calibrated on an annual basis. Post-calibrations showed no drift in the temperature sensor but occasionally some drift in the conductivity. No corrections to the sea surface salinity (SSS) were made and salinity did not undergo further quality control other than removal of spikes and values that were out of range. As salinity has minimal effect on the calculated $fCO_2$, bad salinities were removed and substituted by linearly interpolated values to eliminate gaps. When SST were not recorded in the $CO_2$ files, or in error, the SST gaps were filled from the high resolution

SST data files maintained by the RSMAS Oceanscope project. On the rare occasion that SST was not recorded at all, SST data was estimated from the equilibrator temperature data ($T_{eq}$) after applying a constant offset between $T_{eq}$ and SST using SST data before and after the gap. On average the $T_{eq}$ was $0.12 \pm 0.28$˚C lower than SST for all the cruises but per cruise the standard deviation of $T_{eq}$-SST was on the order of 0.04 ˚C. The in situ $pCO_2$ data calculated with $T_{eq}$ was flagged as questionable.

For the regionally mapped products on a 1-degree grid and monthly timescale (1˚ by 1˚ by mo) SST and SSS were obtained from the following sources: The SSS data were from a numerical model, the Hybrid Coordinate Ocean Model (HYCOM) (https://HYCOM.org/) and referred to as $SSS_{HYCOM}$. The SST product for the whole region is the Optimum Interpolated SST, OISST (Reynolds et al., 2007). It uses data from ships, buoys and satellites to generate the fields. For the OISST the reference SST is from buoys, and the SST obtained from ships were adjusted in the OISST product to the buoy data by subtracting

0.14˚C (Reynolds et al., 2007).

**2.3 Wind speed data**

Winds were measured on the ships but these data are not used as they are not synoptic for the whole region which is a requirement for the regional flux maps. Instead, wind speeds were obtained from the updated cross-calibrated multi-platform wind product (CCMP-2) (Atlas et al., 2011). The mean scalar neutral wind at 10-m height $<u_{10}>$, and its second moment $<u_{10}^2>$

were used to calculate the fluxes. They were determined from the ¼ degree, 6-hourly product that was obtained from Remote Sensing Systems (RSS) (www.remss.com). This product relies heavily on the European Center for Median Weather Forecasting (ECMWF) assimilation scheme that uses in situ and remotely sensed assets, particularly (passive) radiometers on satellites. The directional component uses scatterometer data. The ¼ degree, 6-hourly CCMP-2 product was binned and averaged in 1-degree grid boxes on monthly scales (1˚ by 1˚ by mo). In absence of CCMP-2 data for 2018, the wind product

from European Reanalysis, ERA5, was used (https://www.ecmwf.int/en/forecasts/datasets/reanalysis-datasets/era5) (Copernicus Climate Change Service, 2017). The ERA5 wind data are at 31-km and 3-hourly resolution but were binned and averaged in the same manner as the CCMP-2 winds. There were no apparent biases between the scalar winds for the two products in the Caribbean.

**2.4 Mixed layer depth data**

Mixed layer depths are not necessary to calculate the parameters in the datasets but they are used as an independent variable in the MLRs to interpolate $fCO_{2w}$ values. They are also needed if mixed layer dissolved inorganic carbon (DIC) inventories are desired, and to determine the effect of mixed layer depth on the changes in $fCO_{2w}$. They are therefore provided in the mapped data products. No MLD determinations were made on the cruises and limited observational estimates from other sources are available. The MLDs provided here are from the same numerical model (HYCOM) as used for the mapped SSS

and obtained from http://www.science.oregonstate.edu/ocean.productivity/index.php. The MLDs are based on a density contrast of 0.03 between surface and subsurface, and are only provided in the mapped (1˚ by 1˚ by mo) product.

**3 Calculations**

The calculations of the concentrations and fluxes follow standard procedures as described below. The mapping procedures are detailed, with an example of different possible means of mapping and the effect on the final product.



### 3.1 Calculation of $pCO_2$

The starting point in the calculations, which are aided by MATLAB routines following the procedures as in Pierrot et al. (2009), are calibrated $xCO_2$ values that were referenced against four gas standards covering the range of measurements. Standards were supplied by GMD/ ESRL/NOAA, traceable to the WMO $CO_2$ mole fraction scale.

The $xCO_2$ values were converted to $pCO_{2eq}$ (µatm) values:

$$pCO_{2eq} = xCO_{2eq} (P_{eq}\text{-}pH_2O) \qquad (1)$$

where eq refers to equilibrator conditions. $P_{eq}$ is the pressure in the equilibrator headspace and $pH_2O$ is the water vapor pressure calculated according to Eq. 10 in Weiss and Price (1980). The $pCO_{2eq}$ was corrected to surface water values using the intake temperature (SST) and the temperature of water in the equilibrator ($T_{eq}$) according to the empirical relationship that Takahashi et al. (1993) developed for North Atlantic surface waters:

$$pCO_{2w} = pCO_{2eq}\, e^{(0.0423(SST\text{-}Teq))} \qquad (2)$$

This empirical correction for temperature is widely used but it is of note that using the thermodynamic relationships for carbonate dissociation constants yields different temperature dependencies that are a function of temperature. For average SST in the Caribbean of 27.0 ˚C the temperature dependence varies from 0.036 to 0.040 using commonly used constants as provided in inorganic carbon system programs such as CO2SYS (Pierrot et al, 2006) compared to the coefficient of 0.0423 (or 4.23 % ˚$C^{-1}$) used above. On average SST–$T_{eq}$ = 0.12 ˚C for all the cruises sampled such that the correction from $T_{eq}$ to SST for the coefficient of 0.0423 in Eq. 2 is 1.9 µatm under average conditions of SST= 27 ˚C and $fCO_2$ =374 µatm. Using a temperature coefficient of 0.036 the temperature correction would be 1.6 µatm.

### 3.2 Calculation of $fCO_2$ in air and water

The $fCO_2$ is the $pCO_2$ corrected for non-ideality of $CO_2$ solubility in water using the virial equation of state (Weiss, 1974). The correction can be expressed as $\quad fCO_{2a,w} = e^{g(SST,P)}\, pCO_{2a,w}$

and:

$$g(T,P)=[(\text{-}1636.75+12.0408T\text{-}0.0327957T^2+0.0000316528T^3)+2(1\text{-}xCO_2\, 10^{-6})^2\, (57.7\text{-}0.118T)$$
$$(P/1013.25)]/(82.0575T) \qquad (3)$$

where T is in Kelvin, $xCO_2$ in is ppm, and P is in mbar.

Under average conditions in the Caribbean, the function $e^{g(SST,P)} \approx 0.997$ and $fCO_{2w}$ will be $\approx 1.2$ µatm less than $pCO_{2w}$. As the corrections from partial pressure to fugacity in air and water are approximately the same, the difference between $\Delta pCO_2$ ($=pCO_{2w}\text{-}pCO_{2a}$) and $\Delta fCO_2$ ($=fCO_{2w}\text{-}fCO_{2a}$), that are used to determine the fluxes (Eq. 4), is negligible ($\approx<0.1$ µatm).

### 3.3 Calculation of fluxes

For the determination of the air-sea $CO_2$ flux ($F_{CO2}$, mol $m^{-2}$ $yr^{-1}$), a bulk formulation is applied to the data from the gridded mapped product:



$$F_{CO2} = k \, s \, \Delta fCO_2 \tag{4}$$

where $\Delta fCO_2$ is ($fCO_{2w}$-$fCO_{2a}$), s is the seawater $CO_2$ solubility (Weiss and Price, 1980), and k is the gas transfer velocity parameterized as a function of wind speed (Wanninkhof, 2014):

$$k = 0.251 \, <u_{10}^2> \, (Sc/660)^{-1/2} \tag{5}$$

where $<u_{10}^2>$ is the monthly $2^{nd}$ moment of the wind speeds reported in CCMP-2. The $2^{nd}$ moment accounts for the impact of variability of the wind speed on k. It is determined by taking the monthly average of the sum of squares of the wind speed in CCMP-2 provided at 6-hours and ¼° grid resolution. The number of wind speed observation in a 1° by 1° by mo grid is 1920. This sample density captures the frequency spectrum of winds except that extreme wind events such as hurricanes are not fully represented due to the local nature of the extremes and inherent smoothing in the CCMP-2 product. The Sc is the Schmidt

number of $CO_2$ in seawater, defined as the kinematic viscosity of seawater divided by the molecular diffusion coefficient of $CO_2$. It is determined as a function of temperature from Wanninkhof (2014). At the average temperature of 27 ˚C the $Sc_{CO2}$ equals 475. Over the typical range of SST in the Caribbean from 24 ˚C to 30 ˚C, the $(Sc/660)^{-1/2}$ will vary from 1.1 to 1.27, indicating that the gas transfer velocity will be 27 % higher at an SST of 30 ˚C compared to a SST of 20 ˚C that would correspond to a Sc of 660.

## 3.4 Binning procedure


Binning of the observations of $fCO_{2w}$, SST, and SSS was performed by averaging all the data in each 1° by 1° by mo cell. At typical ship speeds of 22 knots, the ship would cover 1° in about 2.5 hours and take 60 measurements yielding about 250 measurements per month assuming weekly cruises through the area. The actual number of measurements per grid cell ranged from 8 to 500. The higher number of observations per cell were mostly in the latter part of the record, when the *Eqnx* and *AoS*

operated in the same area. The total number of observations from March 2002 through December 2018 is 1.13 million, and the total number of 1° by 1° by mo grid cells with observations is 9224. The standard deviation (stdev) of the $fCO_{2w}$ in each cell is determined and then the average of the stdev for the 9924 cell with observations is taken. The average stdev is 3.4 ± 2.6 µatm (n=9224) indicating the small variability in each cell . The same procedure is followed for SST and SSS yielding values of 0.22± 0.19 ˚C for SST; and 0.10± 0.10 for SSS. These are relatively small deviations compared to the monthly spatial range

of ≈ 20 µatm for $fCO_{2w}$; ≈1 ˚C for SST; ≈ 1 in SSS. The amplitude of the seasonal cycle of ≈ 40 µatm for $fCO_{2w}$ and ≈4 ˚C for SST is significantly greater than the average stdev as well.

The gridding facilitates the co-location and merging of products such as $MLD_{HYCOM}$, $SSS_{HYCOM}$, OISST, $<u^2>_{CCMP}$, and $xCO_{2a,MBL}$ into the gridded observational dataset for further interpretation. The gridded data is also used for comparison of *in situ* SST and SSS with $SSS_{HYCOM}$ and OISST. The average difference between SSS and $SSS_{HYCOM}$ for the 2002-2017 data is -0.1

± 0.28 (n= 8,191), and for SST and OISST the difference is 0.25 ± 0.40 ˚C (n= 8,191) with the in situ SSS being lower and SST being higher. While both differences cover zero within their uncertainty, the temperature difference is in agreement with expected cooler near-surface temperatures that could lead to lower $fCO_{2w}$ which, in turn, has a large impact on the calculated air-sea fluxes.



The gridded observations (1˚ by 1˚ by mo) represent about 10 % of the area of investigation from 15-28 ˚N and 88-62 ˚W over
the period of investigation. To interpolate ("map") the data in space (334 grid cells) and time (192 months) a multi-linear
regression (MLR) approach was used. For each year from 2002 through 2018 the gridded $fCO_{2w}$ observations were regressed
against the 1-degree monthly gridded values of position (Lat and Lon), SST, MLD, and SSS. Other permutations of
independent parameters were tested but yielded less robust fits (see Appendix A). Annual MLRs were created as $fCO_{2w}$ levels
change over time in response to increasing atmospheric $CO_2$ levels. At steady state this increase would raise $fCO_{2w}$ by 2.13
$\mu atm\ yr^{-1}$ over the time period but the observed multi-year (>4-yr intervals) trends varied from -4 to 4 $\mu atm\ yr^{-1}$, with an
average $fCO_{2w}$ trend was 1.4 $\mu atm\ yr^{-1}$ from 2002-2018 (Wanninkhof et al., 2019b).

The annual MLRs have the form of:

280       $fCO_{2w} = a\ Lon + b\ Lat + c\ SST + d\ MLD + e\ SSS + f$                    (6)

The coefficients for each year, along with the error in $fCO_{2w}$ and uncertainties in each of the coefficients are shown in Table
1. The error in the calculated $fCO_{2w}$ ranges from 5 to 9 $\mu atm$. The strongest dependence was with SST. There were significant
cross-correlations between independent variables such that effect of the, often significant, year-to-year differences in
coefficients are difficult to interpret. However, the annual MLRs faithfully reproduce the trends and variability. As illustration,
the results of the MLRs for 2004, 2011, and 2017 (Table 1) were applied to the January data in 2004, 2011, and 2017 and are
plotted in **Figure 3a** along with the observations (if available) for the grid boxes that span the longitude range from 88˚W to
62˚ W between 23˚ N and 24˚ N (see Figure 1). **Figure 3a** shows that the mapped product using annual MLRs reflected the
increases in $fCO_{2w}$ in the region, and consistent differences in patterns between the East and the West for the three years. The
mapped product showed a reasonable correspondence with the gridded observations. Some of the deviations between the
gridded observations and mapped product were caused by the mismatch between SST and SSS *in situ* with the OISST and
$SSS_{HYCOM}$ (**Figures 3b and 3c**). In particular, the strong minima in OISST at 79˚ W were likely due to the OISST capturing
the lower SST near the coast of Cuba. This caused the mapped $fCO_2$ product to be lower as well, as shown in Figure 3a. It
illustrates that the gridded $fCO_2$ product is both influenced by the annual MLR, and the MLD, SST, and SSS products used for
mapping.


### 3.5 Mapping procedures for $fCO_{2w}$ and fluxes

Despite the large number of observations and monthly 1˚ by 1˚gridding, only about 10% of the grid boxes had observations so
some means of temporal and spatial interpolation was necessary. Therefore, the annual MLRs, and modeled and remote sensed
products were used for the mapped products. From the OISST, $MLD_{HYCOM}$, $SSS_{HYCOM}$, that were available for all 1˚ by 1˚ by
mo grid cells, a corresponding $fCO_{2w}$ was determined. For the air-sea $CO_2$ fluxes, the monthly $fCO_{2a}$ values were derived from
the weekly average $xCO_{2a}$ of the stations on Key Biscayne (KEY) and Ragged Point Barbados (RPB) (CarbonTracker Team,

2019; https://www.esrl.noaa.gov/gmd/ccgg/flask.php). The second moments of the scalar winds, $<u^2>$ from CCMP-2 were averaged on the same grids. As no CCMP-2 product was available in 2018 the ERA5 wind product was used for the last year of the record. As noted in Atlas et al. (2011) the ERA5 winds are a part of the ECWFM wind products that are used as a prior

in the CCMP-2, and thus scalar winds between products are very similar.

Table 1. Coefficients for the MLR for each year *

| | a (LON) | b (LAT) | c (SST) | d (MLD) | e (SSS) | f (Icept) | Error $fCO_{2w,MLR}$ | $r^2$ | #points |
|---|---|---|---|---|---|---|---|---|---|
| 2002 | -0.32 | 0.45 | 10.29 | -0.04 | 0.77 | 24.4 | 4.72 | 0.90 | 537 |
| | *0.04* | *0.11* | *0.18* | *0.02* | *0.72* | *25.8* | | | |
| 2003 | -0.56 | 0.32 | 9.24 | -0.09 | 1.11 | 32.6 | 5.21 | 0.86 | 731 |
| | *0.03* | *0.09* | *0.17* | *0.02* | *0.66* | *24.6* | | | |
| 2004 | -0.42 | 0.82 | 10.34 | -0.20 | 2.78 | -55.4 | 5.21 | 0.92 | 740 |
| | *0.03* | *0.09* | *0.17* | *0.01* | *0.63* | *23.6* | | | |
| 2005 | -0.43 | 0.49 | 8.71 | -0.07 | 7.30 | -172.0 | 6.58 | 0.85 | 664 |
| | *0.04* | *0.12* | *0.18* | *0.02* | *0.85* | *32.1* | | | |
| 2006 | -0.31 | 1.13 | 9.60 | -0.19 | 2.56 | -26.6 | 4.99 | 0.89 | 670 |
| | *0.03* | *0.08* | *0.16* | *0.02* | *0.70* | *26.1* | | | |
| 2007 | -0.62 | 1.12 | 10.56 | -0.36 | 2.75 | -77.8 | 7.04 | 0.79 | 483 |
| | *0.05* | *0.15* | *0.32* | *0.03* | *1.33* | *49.0* | | | |
| 2008 | -0.12 | 1.12 | 10.58 | -0.30 | 3.16 | -55.8 | 5.66 | 0.95 | 107 |
| | *0.43* | *0.31* | *0.45* | *0.05* | *1.39* | *59.8* | | | |
| 2009 | -0.66 | 0.30 | 7.18 | -0.47 | 0.11 | 131.9 | 6.91 | 0.84 | 125 |
| | *0.22* | *0.24* | *0.53* | *0.07* | *1.49* | *58.8* | | | |
| 2010 | -0.53 | 2.02 | 8.48 | -0.23 | 2.12 | -10.5 | 9.31 | 0.85 | 323 |
| | *0.17* | *0.26* | *0.41* | *0.05* | *1.48* | *51.7* | | | |
| 2011 | -0.32 | 0.98 | 6.65 | -0.28 | 3.06 | 46.8 | 7.19 | 0.76 | 305 |
| | *0.13* | *0.19* | *0.34* | *0.04* | *1.51* | *59.7* | | | |
| 2012 | -0.13 | 1.66 | 10.81 | -0.33 | 5.56 | -154.1 | 7.29 | 0.91 | 358 |
| | *0.10* | *0.16* | *0.28* | *0.04* | *1.36* | *50.9* | | | |
| 2013 | -0.43 | 1.30 | 11.45 | -0.47 | 4.45 | -137.5 | 8.16 | 0.83 | 219 |
| | *0.18* | *0.23* | *0.48* | *0.05* | *1.63* | *55.5* | | | |
| 2014 | -0.60 | 0.62 | 9.48 | -0.18 | 3.26 | -45.2 | 7.02 | 0.78 | 362 |



| | | | | | | | | | |
|---|---|---|---|---|---|---|---|---|---|
| | *0.08* | *0.15* | *0.29* | *0.03* | *1.06* | *37.7* | | | |
| 2015 | -0.84 | 0.14 | 7.76 | -0.18 | 1.20 | 70.0 | 6.01 | 0.82 | 455 |
| | *0.04* | *0.11* | *0.27* | *0.02* | *0.90* | *34.5* | | | |
| 2016 | -0.79 | 0.37 | 8.72 | -0.14 | 4.17 | -62.7 | 6.66 | 0.82 | 1001 |
| | *0.03* | *0.08* | *0.20* | *0.01* | *0.45* | *17.1* | | | |
| 2017 | -0.42 | 0.56 | 9.49 | -0.22 | 5.65 | -112.6 | 6.17 | 0.87 | 974 |
| | *0.03* | *0.08* | *0.15* | *0.02* | *0.58* | *17.1* | | | |
| 2018 | -0.34 | -0.09 | 10.85 | -0.13 | 6.24 | -151.2 | 5.08 | 0.90 | 1162 |
| | *0.02* | *0.07* | *0.13* | *0.01* | *0.39* | *14.7* | | | |

* These regressions were used to create the mapped $fCO_{2w}$ fields for each year using the 1° by 1° by month gridded data product. $fCO_{2w,Mapped}$ = a Longitude + b Latitude + c SST + d MLD + e SSS + f

The second row (in italics) for each annual entry is the error of the coefficient.

### 3.6 Determination of Alkalinity and mapping of pH

Total Alkalinity (TA) was determined from salinity. Several algorithms have been developed (Millero et al., 1998; Lee et al., 2006; Takahashi et al., 2014 and Cai et al., 2010) that show close agreement for the conditions in the Caribbean. Figure 4 shows several TA-salinity relationships over the salinity range encountered in the region along with error bars for the Cai et al. (2010) relationship depicting the uncertainty of 5.5 µmol $kg^{-1}$. The relationship of Cai et al. (2010), specifically developed for the Caribbean Sea (see insert of Figure 10 in Cai et al., 2010), TA= 57.3SSS+ 296.4, stdev = 5.5 was used here.

As shown in Figure 4 the agreement between relationships was good, and choice of TA relationship did not have a determining influence on results. The calculated TA was used to determine pH with the gridded $fCO_{2w}$ product. The program CO2SYS for excel V2.2 (Pierrot et al., 2006) was used with the apparent $CO_2$ dissociation constants, K1, K2 from Lueker et al. (2000); $KSO_4^-$ dissociation constants from Dickson (1990); KF dissociation constants from Perez and Fraga (1987); and Total Boron salinity relationship from Uppström (1974). The pH product was determined at the OISST and on the total scale ($pH_T$) on the 1° by 1° by mo grid.

This approach of calculating the pH from the monthly product differs from other created pH products (Lauvset et al., 2016; Jiang et al., 2015) where the pH is calculated from the in situ measurements of DIC and TA and then regressed and gridded. To examine the differences derived from using one or the other approach, both were compared for 201. pH was calculated for every $fCO_{2w}$ observation and gridded into 1° x 1° x mo. A MLR was from the calculated pH created for 2017:

$$pH_T(MLR) (\pm 0.005) = 0.0003194\ Lon - 0.00046744\ Lat - 0.00965183\ SST + 0.00019602\ MLD +$$
$$0.00069378\ SSS + 8.3240 \qquad\qquad r^2 = 0.89\ (n = 1244) \qquad (7)$$





This MLR was then applied to the independent variables for each grid box. This was compared to the approach used here of calculating the pH from the mapped $fCO_{2w}$ and TA-SSS relationships on 1˚ x 1˚ x mo grids called $pH_T(fCO_{2w},TA)$.

The two approaches provided similar results with $pH_T(fCO_{2w},TA)$ - $pH_T(MLR)$ = -0.0001 ± 0.005 for 2017. The small difference showed a pattern with SST (**Fig. 5**) but not with the other independent variables. For SST the differences showed a

concave pattern with positive deviations at low SST, a minimum (≈0.000) between 26 ˚C and 28 ˚C, and an increase at higher SST. The differences are insignificant, indicating that the results using $pH_T(fCO_{2w},TA)$ or $pH_T(MLR)$ are very similar such that the approach of using mapped $fCO_{2w}$ and TA to determine $pH_T$, $pH_T$, $(fCO_{2w},TA,)$ yielded precise gridded $pH_T$ .

### 3.7 Aragonite saturation state ($\Omega_{Ar}$)

The aragonite saturation state ($\Omega_{Ar}$) indicates the level of supersaturation, or undersaturation of seawater with respect to the

mineral aragonite, a polymorph of calcium carbonate, and part of the skeletal structure of many marine calcifiers. Values of $\Omega_{Ar}$ that are less than one indicate that aragonite would dissolve, and greater than one it would have a tendency to precipitate. It is used as an indicator of ecosystems health with regards to ocean acidification. In warm tropical regions saturation states are well above one but no active precipitation takes place except under unusual circumstances in shallow waters, in a precipitation process called whitings (Purkis et al., 2017). The $\Omega_{Ar}$ is not measured directly and is defined as the product of

calcium and carbonate ion concentrations divided by the solubility product of aragonite:

$$\Omega_{Ar} = [Ca^{2+}] [CO_3^{2-}] (K_{Ar}'_{sp})^{-1} \qquad (8)$$

where $[Ca^{2+}]$ is the total calcium concentration and is derived from salinity: $[Ca^{2+}]$ =293.86 S (Millero 1995). $[CO_3^{2-}]$ is the

total carbonate ion concentration determined from two of the four measured inorganic carbon system parameters, and $K_{Ar}'_{sp}$ is the apparent solubility product of aragonite in seawater at a specified salinity, temperature and pressure. In this work $[CO_3^{2-}]$ was determined from the monthly binned $fCO_{2w}$ and TA using the CO2SYS program. For surface waters, $K_{Ar}'_{sp}$ is

$$pK_{Ar}'_{sp} = -[-171.945 - 0.077993\ T + 2903.293/T + 71.595log\ (T) + (-0.068393 + 0.0017276\ T +$$
$$88.135/T)\ S^{0.5} - 0.10018\ S + 0.0059415\ A1^{.5}] \qquad (9)$$

Where $pK_{Ar}'_{sp}$ = -log $K_{Ar}'_{sp}$ , T is temperature in Kelvin (K), and S is salinity (Mucci,1983). As with $pH_T$, the $\Omega_{Ar}$ product was determined from the mapped (1˚ by 1˚ by mo) values of SSS, SST, $fCO_{2w}$ and TA.

### 4 Datasets and data products

Several different data products are provided in conjunction with this paper. The methodology to create the products is described above, and here the file format and column headers is presented with a brief description when warranted.



## 4.1 Underway pCO₂ data

The quality controlled cruise data is posted at different locations. The data file structure is from the MATLAB data reduction program of Pierrot (pers. comm.). The individual cruise files with metadata can be found at
https://www.aoml.noaa.gov/ocd/ocdweb/occ.html. Data can be found as part of the SOCAT holdings (Bakker et al. 2016) using an interactive graphical user interface https://ferret.pmel.noaa.gov/socat/las/. In addition, cruise files of the three ships are provided in annual folders at the National Center for Environmental Information (NCEI) (https://www.nodc.noaa.gov/ocads/oceans/VOS_Program/explorer.html). The primary identifier for the cruises is the EXPO code which is the International Council for the Exploration of the Sea (ICES) ship code and the day the ship starts the cruise.
Examples are as follows: For a cruise of the *EoS* starting March 6, 2002, the EXPO code is 33KF20020330; for a *AoS* cruise starting November 25, 2018, the EXPO code is BHAF20181125, and for the *Eqnx* cruise departing her homeport on February 10, 2018, it is MLCE20180210. The individual cruise files sometimes include data outside the study region.

## 4.2 Gridded data

The gridded datasets are the averaged fCO₂w, SST, and SSS observation on 1˚ by 1˚ by mo grid. The files include the auxiliary
data obtained from remote sensing and interpolated data (OISST), data assimilation of remotely sensed winds (CCMP-2) and from the HYCOM model (SSS$_{HYCOM}$). Calculated TA, pH$_T$ and Ω$_{Ar}$ using procedures outlined above are provided in the file. The calculated fCO₂w using the annual MLRs (Table 1) are provided as well. This data set has spatial and temporal gaps as the ships did not transit through each pixel, and coverage is uneven. The number of observations differ for each grid point and are included in the gridded data set. For the auxiliary data the number of data points are fixed by the resolution of the data products
except where part of the grid includes land which is masked. The column headers are provided in Table 2 and include units and descriptions when warranted.

Table 2. Column headers for the monthly 1-degree gridded observational product (1˚ by 1˚ by mo)

| Parameter | Unit | Description |
|---|---|---|
| Year | | |
| Month | | 1 (January) through 12 (December) |
| Lat | Degrees | North is positive. Location is the center point of the grid. That is, 15.5˚ N is the grid box spanning 15˚ N to 16˚ N |
| Lon | Degrees | East is positive. All values in the Caribbean are negative. Location is center point of grid. That is, -87.5 is the grid box spanning 87˚ W to 88˚ W |
| Area | Km$^2$ | Area of grid box excluding land where appropriate |
| #_Obs | | Number of fCO₂w observations in the particular grid box for the particular month |
| SST_OBS | ˚C | Sea surface temperature measured at the intake (average of the grid box) |





| SST_STD | °C | Standard deviation of SST |
|---|---|---|
| SSS_OBS | permil | Sea surface salinity measured by thermosalinograph (average of the grid box) |
| SSS_STD | | Standard deviation of SSS |
| $fCO_{2w}$_OBS | µatm | Fugacity of $CO_2$ in seawater (average of the grid box) |
| $fCO_{2w}$_STDEV | | Standard deviation of $fCO_{2w}$ observations in the grid box |
| TA | µmol kg$^{-1}$ | Total alkalinity calculated from a relationship salinity TA = 57.3SSS_OBS+ 296.4 (Cai et al., 2010) using the measured SSS |
| $pH_T$ | | pH on the total scale at SST calculated from $fCO_{2w}$_OBS and TA using the CO2SYS program of Pierrot et al. (2006) with pH Scale: Total scale (mol/kg-SW) at OISST; $CO_2$ Constants: K1, K2 from Lueker et al. (2000); $KSO_4^{-}$ for Dickson (1990); KF from Perez and Fraga (1987) and Total Boron from Uppström (1974) |
| $\Omega_{Ar}$ | | Aragonite saturation state calculated using CO2SYS with $fCO_{2w}$_OBS and TA as input parameters and the same dissociation constants as used for $pH_T$ |
| OISST | °C | Optimal interpolated sea surface temperature (Reynolds et al., 2007) for the particular grid box |
| $SSS_{HYCOM}$ | permil | Sea surface salinity from the HYCOM model |
| $fCO_{2w}$_MLR | µatm | Fugacity of $CO_2$ in seawater determined from annual MLRs (see Table 1) with Lat, Lon, SST, OISST, $SSS_{HYCOM}$, and $MLD_{HYCOM}$. |






The comparison between observed SST and OISST showed that the OISST was on average 0.25 ˚C lower than SST. The SSS$_{HYCOM}$ was 0.10 higher than SSS. The average difference between fCO$_{2w}$_OBS and fCO$_{2wMLR}$ was 1.5 μatm with the fCO$_{2w}$_OBS being higher. While the differences are within the standard deviation, they are possibly real due to known near-surface gradients in the region. No attempt was made to normalize the SST and SSS to OISST and SSS$_{HYCOM}$.


### 4.3 Mapped data product

The mapped data product extends the observations through gap filling of missing 1˚ by 1˚ by mo grid boxes utilizing the annual MLRs of fCO$_{2w}$ as a function of Lat, Lon, OISST, SSS$_{HYCOM}$, and MLD$_{HYCOM}$ for the region from 15˚ N to 28˚ N and -62˚ to -88˚ (= 62˚ W to 88˚ W). The modeled and remotely sensed products were used as the independent parameters: OISST,

SSS$_{HYCOM}$, and MLD$_{HYCOM}$. The mapped product includes the air-sea CO$_2$ fluxes in the region as a specific flux (mol m$^{-2}$ yr$^{-1}$) for each grid box. As noted above, the ΔfCO$_2$ using OISST compared to SST_OBS was 1.5 μatm smaller leading to a larger average CO$_2$ flux into the ocean. Wanninkhof et al. (2019b) showed a 27 % decrease in average specific uptake of CO$_2$ from -0.87 to -0.63 mol m$^{-2}$ yr$^{-1}$ if SST rather than OISST was used for the 2002-2017 time period. The column headers are provided in Table 3 including units and descriptions when warranted.


Table 3. Column headers for the monthly 1-degree mapped product (1˚ by 1˚ by mo) for the whole region

| Parameter | Unit | Description |
|---|---|---|
| Year | | |
| Month | | 1 (January) through 12(December) |
| Lat | Degrees | North is positive. Location is center point of grid. For example, 15.5˚ N is the grid box spanning 15˚ N to 16˚ N |
| Lon | Degrees | East is positive. All values in Caribbean are negative. Location is center point of grid. For example, -87.5 is the grid box spanning 87˚ W to 88˚ W |
| Area | Km$^2$ | Area of grid box excluding land where appropriate |
| OISST | ˚C | Optimal interpolated sea surface temperature (Reynolds et al., 2007) |
| SSS$_{HYCOM}$ | permil | Sea surface salinity Sea surface salinity from HYCOM |
| MLD$_{HYCOM}$ | m | Mixed layer depth from the HYCOM model |
| fCO$_{2wMLR}$ | μatm | Fugacity of CO$_2$ in seawater determined from annual MLRs with Lat, Lon, SST, SSS, and MLD (see Table 1) |
| fCO$_{2a}$ | μatm | Fugacity of CO$_2$ in air using the average value between atmospheric sampling station KEY and RBP |
| ΔfCO$_2$ | μatm | Air water fugacity difference, fCO$_{2w}$ - fCO$_{2a}$ |





| TA | μmol kg$^{-1}$ | Total alkalinity calculated from a relationship salinity TA = 57.3SSS$_{HYCOM}$+ 296.4 (Cai et al., 2010) |
|---|---|---|
| pH$_T$ | | pH calculated from fCO$_{2w}$ and TA with the CO2SYS program of Pierrot et al. (2006) with pH Scale: Total scale (mol/kg-SW) at OISST; CO$_2$ Constants: K1, K2 from Lueker et al. (2000); KSO$_4^-$ for Dickson (1990); KF from Perez and Fraga (1987) and Total Boron from Uppström (1974) |
| Ω$_{Ar}$ | | Aragonite saturation state calculated using CO2SYS with fCO$_{2wMLR}$, TA, OISST, and SSS$_{HYCOM}$ as input parameters with same dissociation constants as used for pH$_T$ |
| <u$^2$> | m$^2$ s$^{-2}$ | Second moment of the wind based on ¼° 6-h CCMP-2 product (Atlas et al., 2011) |
| CO$_2$_Flux | mol m$^{-2}$ yr$^{-1}$ | Monthly air-sea CO$_2$ flux calculated according to Eqns. 4 and 5 |

## 4.5 Monthly and Annual estimates for the Caribbean 2002-2018

Summary files of monthly and annual data and products covering the whole region from 15º N to 28˚ N and 88º W and 62˚ W

are provided based on summing the data in the mapped products. The column headers for the monthly and annual products are similar, and provided in Tables 4 and 5. The standard deviations of the monthly files represent the spatial deviations of the sums and averages for the monthly grid boxes for the region. For the annual files, the standard deviation (stdev) is the temporal deviation based on the monthly averages (n=12). The averages were weighted according to area of each 1˚ by 1˚grid box. The total CO$_2$ mass flux (CO$_2$_Flux$_{Total}$) is the integral of the monthly CO$_2$ fluxes (mol m$^{-2}$ yr$^{-1}$) expressed in tera-grams (= 10$^{12}$ g)

per month or per year.

Of note is that the quantities, such as TA, pH$_T$, Ω$_{Ar}$ and CO$_2$-Flux, are not calculated from the variables presented as monthly or annual parameters but rather they are sums of the monthly grid boxes. For example, the monthly average TA is the area weighted average of the TA calculated for each grid box which will not be the same as using the parameterization, TA=57.3SSS$_{HYCOM}$+ 296.4 and the monthly averaged SSS$_{HYCOM}$.


Table 4. Column headers for the monthly averaged mapped product

| Parameter | Unit | Description |
|---|---|---|
| Year | | |
| Month | | 1 (January) through 12 (December) |
| Area | km$^2$ | Total area of Caribbean region excluding land (15˚ N to 28˚ N and 62˚ W to 88˚ W) |
| OISST | ˚C | Optimal interpolated sea surface temperature (Reynolds et al., 2007) |



| $SSS_{HYCOM}$ | permil | Sea surface salinity Sea surface salinity from HYCOM |
|---|---|---|
| $MLD_{HYCOM}$ | m | Mixed layer depth from the HYCOM model |
| $fCO_{2wMLR}$ | µatm | Fugacity of $CO_2$ in seawater determined from annual MLRs with Lat, Lon, SST, SSS, and MLD (see Table 1) |
| $fCO_{2a}$ | µatm | Fugacity of $CO_2$ in air using the average value between atmospheric sampling station KEY and RBP |
| $\Delta fCO_2$ | µatm | Air-water fugacity difference, $fCO_{2w}$ - $fCO_{2a}$ |
| TA | µmol $kg^{-1}$ | Total alkalinity calculated from a relationship salinity TA = $57.3SSS_{HYCOM}$+ 296.4 (Cai et al., 2010) |
| $pH_T$ | | pH calculated from $fCO_{2w}$ and TA with the CO2SYS program of Pierrot et al. (2006) with pH Scale: Total scale (mol/kg-SW) at OISST; $CO_2$ Constants: K1, K2 from Lueker et al. (2000); $KSO_4^-$ for Dickson (1990); KF from Perez and Fraga (1987) and Total Boron from Uppström (1974) |
| $\Omega_{Ar}$ | | Aragonite saturation state calculated using CO2SYS with $fCO_{2wMLR}$, TA, OISST, and $SSS_{HYCOM}$ as input parameters with same dissociation constants as used for $pH_T$ |
| $<u^2>$ | $m^2 s^{-2}$ | Second moment of the wind based on ¼° 6-h CCMP-2 product (Atlas et al., 2011) |
| $CO_2\_Flux$ | mol $m^{-2} mo^{-1}$ | Monthly air-sea $CO_2$ flux calculated according to Eqns. 4 and 5 |
| $CO_2\_Flux_{Total}$ | Tg C $mo^{-1}$ | Total monthly air-sea $CO_2$ flux calculated according to Eqns. 4 and 5 in Teragram carbon |

Table 5. Column headers for the annual averaged mapped product

| Year | | |
|---|---|---|
| Area | $km^2$ | Total area of Caribbean region, excluding land, from 15° N to 28° N and 62° W to 88° W |
| OISST | °C | Optimal interpolated sea surface temperature (Reynolds et al., 2007) |
| $SSS_{HYCOM}$ | permil | Sea surface salinity Sea surface salinity from HYCOM |
| $MLD_{HYCOM}$ | m | Mixed layer depth from the HYCOM model |
| Area | $Km^2$ | Area of grid box excluding the surface area of land where appropriate |
| $fCO_{2wMLR}$ | µatm | Fugacity of $CO_2$ in seawater determined from annual MLRs with Lat, Lon, SST, SSS, and MLD (see Table 1) |
| $fCO_{2a}$ | µatm | Fugacity of $CO_2$ in air using the average value between atmospheric sampling station KEY and RBP |





| $\Delta fCO_2$ | $\mu$atm | Air water fugacity difference, $fCO_{2w}$ - $fCO_{2a}$ |
|---|---|---|
| TA | $\mu$mol kg$^{-1}$ | Total alkalinity calculated from a relationship salinity TA = 57.3SSS$_{HYCOM}$+ 296.4 (Cai et al., 2010) |
| pH$_T$ | | pH calculated from $fCO_{2w}$ and TA with the CO2SYS program of Pierrot et al. (2006) with pH Scale: Total scale (mol/kg-SW) at OISST; CO$_2$ Constants: K1, K2 from Lueker et al. (2000); KSO$_4^-$ for Dickson (1990); KF from Perez and Fraga (1987) and Total Boron from Uppström (1974) |
| $\Omega_{Ar}$ | | Aragonite saturation state calculated using CO2SYS with $fCO_{2wMLR}$, TA, OISST, and SSS$_{HYCOM}$ as input parameters with same dissociation constants as used for pH$_T$ |
| $<u^2>$ | m$^2$ s$^{-2}$ | Second moment of the wind based on ¼° 6-h CCMP-2 product (Atlas et al., 2011) |
| CO$_2$_Flux | mol m$^{-2}$ yr$^{-1}$ | Annual air-sea CO$_2$ flux calculated according to Eqns. 4 and 5 |
| CO$_2$_Flux$_{Total}$ | Tg y$^{-1}$ | Total annual air-sea CO$_2$ flux calculated according to Eqns. 4 and 5 in Tera-gram carbon |

**5 Data availability**

The observations are available at three locations in slightly different formats but all files are stored by ship and cruise. The primary source is the website at the Atlantic Oceanographic and Meteorological Laboratory (AOML) (https://www.aoml.noaa.gov/ocd/ocdweb/occ.html). These data are submitted to SOCAT at least once a year such that they can be posted in the annual updates of SOCAT (https://socat.info). The permanent depository of the data is at NCEI where the

data are stored per cruise in directories listed per year (https://www.nodc.noaa.gov/ocads/oceans/VOS_Program/explorer.html). The gridded observations and mapped products described herein are posted at directories at AOML and NCEI The dataset, and derived quantities are provided on a 1-degree monthly grid at http://accession.nodc.noaa.gov/0207749, DOI:10.25921/2swk-9w56 (Wanninkhof et al., 2019a). The products cover the years 2002 through 2018 and will be updated annually.

**6 Conclusions**

The dataset from the cruise ships sailing the Caribbean Sea is a rich resource for studying the inorganic carbon cycling and ocean acidification in the region. The scales of variability and data density are such that 1 degree monthly binning captures the magnitudes and trends of $fCO_{2w}$ and derived inorganic carbon products on seasonal to interannual scales. Using annual MLRs to interpolate $fCO_{2w}$ with position, SST, SSS, and MLD as independent variables yielded accurate monthly products

(Wanninkhof et al., 2019a). A comprehensive investigation of the changes in decadal trends based on the dataset and products was presented in Wanninkhof et al. (2019b). The $fCO_{2w}$ observations, and gridded and mapped products including TA, $pH_T$ and $\Omega_{Ar}$ are used to determine the artifacts and biases that can occur from extrapolation of data. The MLRs capture the spatial and temporal variability in $fCO_{2w}$ and calculated $pH_T$ and $\Omega_{Ar}$ well in the region. The datasets and products are invaluable for model initiation and validation, and serve as boundary conditions for near-shore fine scale models.

**Team list**

This work would not have been possible without support from Royal Caribbean Cruise Lines (RCCL) who have provided access to their ships and significant financial, personnel and infrastructure resources for the measurement campaign coordinated through the Rosenstiel School of Marine and Atmospheric Sciences (RSMAS) of the University of Miami (U. Miami). Peter Ortner, Elizabeth Williams, Don Cucchiara and Chip Maxwell of the Marine Technical group at RSMAS/U.

Miami have been instrumental in keeping the science operations going. Denis Pierrot, Kevin Sullivan, Leticia Barbero, Robert Castle (ret.), and Betty Huss of NOAA AOML have led the gathering, maintenance, data processing and posting of $fCO_2$ data. In addition to $fCO_2$ measurements; skin temperature (MAERI) led by P. Minnet and M. Izaguirre, RSMAS; TSG with instruments supplied by G. Goni and F. Bringas of AOML; optics, and ADCP, with data processed at the University of Hawaii Currents center (E. Firing and J. Hummon) operations take place on the RCCL ships.

**Author contributions**

All authors contributed to writing and editing the documents. JT performed most of the gridding and binning, and provided the model and remotely sensed data from the sources listed in the text. KF and DP performed maintenance and data reduction, and liaised with all parties involved in the operations.

**Competing interests**

The authors of this manuscript have no competing interest involving this work.

**Acknowledgments**

David Munro INSTAR, ESRL/GMD provided the KEY and RPB $CO_2$ data. NOAA Optimal Interpolated SST data were provided by the NOAA/OAR/ESRL PSD through https://www.esrl.noaa.gov/psd/. The NOAA office of oceanic and atmospheric research (OAR) is acknowledged for financial support, in particular the Ocean Observations and Monitoring

Division (OOMD) (fund reference 100007298), and the NOAA/OAR Ocean Acidification Program.



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






**Figure captions**

Figure 1. Map with the cruise tracks of the *EoS*, *Eqnx*, and *AoS* from 2002 though 2018. The green rectangle depicts the region where the data and products are compared in Figure 3.

Figure 2. Histogram of number of cruises per year used in this work.

Figure 3. Zonal section of mapped and gridded data products between 23˚ N and 24˚ N for January 2004 (Black), January 2011 (Blue), and January 2017 (Red). a. $fCO_{2w}$; b. SST; c. SSS. The lines with small solid circles are the mapped product, the larger solid circles are the gridded data with standard deviation of data in the box shown as error bars.

Figure 4. Relationships of surface water TA with salinity. The relationship of Cai et al. (2010) with uncertainty shown as error
bars is used to calculate pH and $\Omega_{Ar}$ from $fCO_{2w}$.

Figure 5. pH calculated from gridded $fCO_{2w}$ and TA estimated from SSS, $pH_T(fCO_2,TA)$, as done in this work, minus pH calculated from observed $fCO_{2w}$ and TA from SSS, $pH_T(MLR)$ plotted against temperature.





Appendix A

# A 17-year dataset of surface water fugacity of CO₂, along with calculated pH, Aragonite saturation state, and air-sea CO₂ fluxes in the Caribbean Sea


Rik Wanninkhof, Denis Pierrot, Kevin Sullivan, Leticia Barbero, and Joaquin Triñanes

Multi-linear regressions (MLRs) for $fCO_{2w}$ were applied to the data on a 1˚ x 1˚ x mo grid using sequentially fewer independent parameters, and the increase in residual was determined. This analysis was performed as two of the independent parameters

used, SSS and MLD, are modeled and their accuracy is not readily known. The functional form for the multi-linear regression, MLR fit is:

$$fCO_{2w,MLR} = a\ Lon + b\ Lat + c\ SST + [d\ MLD] + [e\ SSS] + f \tag{A1}$$

where the full number of independent parameters are used in the data products described the main text and the terms in brackets indicate the parameters omitted in the estimates here. The coefficients for the MLRs and their standard errors for each year without MLD, and the MLRs without MLD and SSS, are provided in **Tables A1 and A2**. SST and location are the strongest predictors of $fCO_{2w}$ levels in the region. The increase in the average root mean square of the residual (RMSE) in Tables A1 and A2 by excluding MLD and SSS in the annual MLRs is relatively small. The RMSE of the estimate $fCO_{2w,MLR}$ increases by

$8 \pm 5.5$ % by excluding MLD, and increases by $11 \pm 5.4$ % when MLD and SSS are omitted with the annual differences provided in the last column of Tables A1 and A2.

SST is the strongest predictor for the MLRs for all the years. Olsen et al. (2004) showed a strong correlation between $fCO_{2w}$, SST and location for the study area as well in 2003. Other observations in subtropical and tropical oceans show a similar strong

correlation between $fCO_{2w}$ and SST (e.g. Lefèvre et al. 2014; Bates et al, 2014). Including SSS and MLD improves the multi-linear regression used for spatial gap filling to some extent (Tables A1-A2). The MLD is obtained from a numerical model, as no full observational record of mixed layer depths is available for the Caribbean. The residuals of $fCO_{2w,MLR}$ versus binned $fCO_{2w}$ for grid boxes with $fCO_{2w}$ observations show no spatial or seasonal pattern with any of the combinations of independent variables used in Eq. A1.


For the entire record, from 2002-2018, single regressions of $fCO_{2w}$ with position, SST, SSS, and MLD showed larger uncertainty as they do not capture the changes in $fCO_{2w}$ over time due to $fCO_{2w}$ increases in response to increasing atmospheric





$CO_2$ levels. Regressing against $\Delta fCO_2$ which, in principle should not have a trend over time if surface water $CO_2$ levels keep up with atmospheric $CO_2$ increases, did not improve the correlation with the independent parameters. This was attributed to
the relatively small magnitude of $\Delta fCO_2$ and local scale variability in $\Delta fCO_2$ that is not strongly linearly correlated with the independent variables. However, using the year as a variable in the regression provides a reasonable means of extrapolating over the entire time/space domain with a single regression from 3/2002 through 2/2018:

$fCO_{2w,MLR}$ (±7.8) = 23.3 (±2.0) +1.45 (±0.02) (YR-2002)+ 10.23 (±0.06) SST +
670              1.19 (±0.03) Lat – 0.50 (±0.01) Lon,              $r^2 = 0.84$              (A2)

where YR is the integer calendar year. Thus YR-2002 is the year since the start of the record. The coefficient for the (YR-2002) term of 1.46 reflects the annual increase in surface water $fCO_{2w}$ due to atmospheric increase, which averages 2.3 ppm $yr^{-1}$ over the 2002-2018 time period.

The overall uncertainty in the $fCO_{2w,MLR}$ of 7.8 µatm in Eq. A2 is generally greater than the uncertainty in the annual algorithms used to fill the gaps for the annual estimates (Eq. A1) that range from 4.7 to 9.9 µatm depending on the year (Table 1 in main text).

The importance of the different independent variables for the $fCO_{2w,MLR}$ can, in part, be discerned from the uncertainty in the coefficients but since variables are cross –correlated, other means are investigated such as creating a MLR with either a subset or substitution of variables. Physical parameters were correlated with location (Lat, Lon) in the region. In particular, salinity showed broad correspondence with position. Therefore, substituting SSS for Lat and Lon provided similar magnitude and uncertainty in the coefficients of the independent variable, but a 10 % greater RMS in the estimated $fCO_{2w}$:

$fCO_{2w,MLR}$ (±8.8) = -238 (±9.8) +1.36 (±0.02) (YR-2002)+ 10.11 (±0.06) SST +
              9.07 (±0.25) SSS,              $r^2 = 0.79$              (A3)

Finally, a simple two parameter linear fit with YR and SST had reasonable predictive capability:

$fCO_{2w,MLR}$ (±9.4) = 107.2 (±1.7) +1.38 (±0.02) (YR-2002)+ 9.50 (±0.06) SST              $r^2 = 0.77$              (A4)

Eq. A4 which used only SST and time showed a small increase in the uncertainty of the derived independent variable $fCO_{2w}$
compared to the other permutations of the MLR. This simple relationship did show some biases with location (not shown),



and for gap filling to create uniform monthly fields of fCO$_{2w}$ the full annual regressions (Table 1) using all independent parameters, SST, SSS, MLD and location was the best option.

Table A1. Coefficients for the MLR, fCO$_{2w,MLR}$ = a Longitude + b Latitude + c SST + e SSS + f


| | a (LON) | b (LAT) | c (SST) | f (Icept) | RMSE fCO$_{2w,MLR}$ | r$^2$ | #points | % increase RMSE** |
|---|---|---|---|---|---|---|---|---|
| 2002 | -0.30 | 0.48 | 10.42 | 16.6 | 4.74 | 0.90 | 537 | 0.4 |
| * | 0.04 | 0.11 | 0.17 | 25.7 | | | | |
| 2003 | -0.53 | 0.35 | 9.60 | 26.7 | 5.32 | 0.86 | 731 | 2.1 |
| | 0.03 | 0.09 | 0.16 | 25.1 | | | | |
| 2004 | -0.35 | 0.89 | 11.52 | -118.3 | 5.79 | 0.90 | 740 | 11.1 |
| | 0.03 | 0.10 | 0.16 | 25.7 | | | | |
| 2005 | -0.41 | 0.53 | 9.08 | -198.3 | 6.65 | 0.85 | 664 | 1.1 |
| | 0.04 | 0.12 | 0.15 | 31.8 | | | | |
| 2006 | -0.20 | 1.26 | 10.23 | -98.5 | 5.48 | 0.87 | 670 | 9.8 |
| | 0.03 | 0.09 | 0.16 | 27.9 | | | | |
| 2007 | -0.50 | 1.33 | 12.05 | -246.0 | 7.96 | 0.74 | 483 | 13.1 |
| | 0.05 | 0.17 | 0.33 | 53.0 | | | | |
| 2008 | 0.26 | 1.77 | 12.50 | -138.3 | 6.43 | 0.94 | 107 | 13.6 |
| | 0.49 | 0.33 | 0.32 | 65.9 | | | | |
| 2009 | -0.38 | 0.24 | 9.26 | -7.8 | 7.95 | 0.79 | 125 | 15.1 |
| | 0.25 | 0.28 | 0.48 | 62.8 | | | | |
| 2010 | -0.45 | 2.43 | 10.11 | -61.4 | 9.66 | 0.84 | 323 | 3.8 |
| | 0.18 | 0.26 | 0.25 | 52.6 | | | | |
| 2011 | -0.49 | 1.03 | 8.11 | -157.1 | 7.83 | 0.71 | 305 | 8.9 |
| | 0.14 | 0.20 | 0.31 | 58.0 | | | | |
| 2012 | -0.19 | 1.95 | 12.27 | -253.6 | 8.01 | 0.89 | 358 | 9.9 |
| | 0.11 | 0.17 | 0.24 | 54.5 | | | | |
| 2013 | -0.13 | 1.89 | 13.04 | -256.5 | 9.66 | 0.76 | 219 | 18.4 |
| | 0.21 | 0.27 | 0.53 | 63.9 | | | | |
| 2014 | -0.31 | 0.96 | 9.89 | -20.1 | 7.40 | 0.76 | 362 | 5.4 |





|  |  |  |  |  |  |  |  |  |
|---|---|---|---|---|---|---|---|---|
|  | *0.07* | *0.15* | *0.30* | *39.5* |  |  |  |  |
| 2015 | -0.86 | 0.09 | 7.96 | 23.8 | 6.65 | 0.78 | 455 | 10.6 |
|  | *0.04* | *0.12* | *0.30* | *37.8* |  |  |  |  |
| 2016 | -0.75 | 0.32 | 9.57 | -121.2 | 6.95 | 0.80 | 1001 | 4.4 |
|  | *0.03* | *0.09* | *0.18* | *16.6* |  |  |  |  |
| 2017 | -0.42 | 0.56 | 9.49 | -112 | 6.16 | 0.87 | 969 | 0 |
|  | *0.03* | *0.09* | *0.15* | *21.3* |  |  |  |  |
| 2018 | -0.34 | 0.009 | 11.57 | -173.2 | 5.35 | 0.89 | 1164 | 5.3 |
|  | *0.02* | *0.07* | *0.12* | *15.43* |  |  |  |  |

*The second row (in italics) for each annual entry is the error of the coefficient.

** The increase in root mean square error (RMSE) of $fCO_{2w,MLR}$ compared to Table 1 that includes MLD as an input

Table A2. Coefficients for the MLR, $fCO_{2w,MLR}$ = a Longitude + b Latitude + c SST + f


|  | a (LON) | b (LAT) | c (SST) | f (Icept) | RSME $fCO_{2w,MLR}$ | $r^2$ | #points | % increase RMSE** |
|---|---|---|---|---|---|---|---|---|
| 2002 | -0.28 | 0.53 | 10.45 | 47.1 | 4.74 | 0.90 | 537 | 0.4 |
| * | *0.04* | *0.10* | *0.17* | *5.3* |  |  |  |  |
| 2003 | -0.54 | 0.40 | 9.56 | 60.9 | 5.33 | 0.86 | 731 | 2.3 |
|  | *0.03* | *0.09* | *0.16* | *5.2* |  |  |  |  |
| 2004 | -0.37 | 1.14 | 11.30 | 9.4 | 5.89 | 0.90 | 740 | 13.1 |
|  | *0.03* | *0.09* | *0.15* | *5.0* |  |  |  |  |
| 2005 | -0.35 | 1.08 | 8.68 | 84.0 | 7.05 | 0.83 | 664 | 7.1 |
|  | *0.04* | *0.11* | *0.16* | *5.9* |  |  |  |  |
| 2006 | -0.18 | 1.56 | 10.06 | 47.8 | 5.59 | 0.86 | 670 | 12.0 |
|  | *0.03* | *0.07* | *0.16* | *6.4* |  |  |  |  |
| 2007 | -0.52 | 1.77 | 11.87 | -31.8 | 8.10 | 0.73 | 483 | 15.1 |
|  | *0.05* | *0.13* | *0.34* | *12.8* |  |  |  |  |
| 2008 | 0.46 | 2.17 | 12.42 | 11.9 | 6.50 | 0.93 | 107 | 14.8 |
|  | *0.49* | *0.30* | *0.33* | *28.4* |  |  |  |  |
| 2009 | -0.42 | 0.44 | 9.04 | 82.0 | 7.99 | 0.78 | 125 | 15.6 |
|  | *0.25* | *0.24* | *0.46* | *20.3* |  |  |  |  |



| 2010 | -0.45 | 2.67 | 10.11 | 3.8 | 9.67 | 0.84 | 323 | 3.9 |
|---|---|---|---|---|---|---|---|---|
|  | *0.18* | *0.18* | *0.25* | *12.4* |  |  |  |  |
| 2011 | -0.37 | 1.65 | 7.76 | 101.1 | 8.09 | 0.69 | 305 | 12.5 |
|  | *0.15* | *0.16* | *0.31* | *14.9* |  |  |  |  |
| 2012 | -0.18 | 2.42 | 11.95 | -13.8 | 8.22 | 0.88 | 358 | 12.8 |
|  | *0.12* | *0.14* | *0.23* | *11.0* |  |  |  |  |
| 2013 | -0.34 | 2.39 | 13.27 | -60.7 | 9.88 | 0.75 | 219 | 21.1 |
|  | *0.20* | *0.22* | *0.53* | *23.5* |  |  |  |  |
| 2014 | -0.32 | 1.17 | 9.90 | 62.9 | 7.44 | 0.75 | 362 | 6.0 |
|  | *0.07* | *0.12* | *0.30* | *11.6* |  |  |  |  |
| 2015 | -0.86 | 0.22 | 7.83 | 100.7 | 6.68 | 0.77 | 455 | 11.1 |
|  | *0.04* | *0.11* | *0.29* | *9.7* |  |  |  |  |
| 2016 | -0.70 | 0.67 | 9.84 | 52.3 | 7.37 | 0.77 | 1001 | 10.7 |
|  | *0.04* | *0.08* | *0.19* | *6.3* |  |  |  |  |
| 2017 | -0.40 | 1.07 | 10.33 | 52.8 | 7.02 | 0.83 | 967 | 13.8 |
|  | *0.03* | *0.08* | *0.16* | *5.32* |  |  |  |  |
| 2018 | *-0.37* | *0.48* | *11.22* | *44.94* | 5.82 | 0.87 | 1165 | 14.6 |
|  | *0.026* | *0.07* | *0.13* | *4.23* |  |  |  |  |

\* The second row (in italics) for each annual entry is the error of the coefficient.

\*\* The increase in root mean square error (RMSE) of $fCO_{2w,MLR}$ compared to Table 1 that included MLD and SSS as an independent variable.

This is the same as Table A1 except that MLD in addition to SSS are omitted as independent variables.

**References Appendix A**

Bates, N. R., Astor, Y. M., Church, M. J., Currie, K., Dore, J. E., González-Dávila, M., Lorenzoni, L., Muller-Karger, F., Olafsson, J., and Santana-Casiano, J. M.: A time-series view of changing ocean chemistry due to ocean uptake of anthropogenic $CO_2$ and ocean acidification., Oceanography, 27, 126-141, http://dx.doi.org/10.5670/oceanog.2014.16., 2014.


Lefèvre, N., Urbano Domingos, F., Gallois, F., and Diverrès, D.: Impact of physical processes on the seasonal distribution of the fugacity of $CO_2$ in the western tropical Atlantic, Journal of Geophysical Research: Oceans, 119, 646-663, 10.1002/2013JC009248, 2014.



Olsen, A., Triñanes, J., and Wanninkhof, R.: Sea-air flux of $CO_2$ in the Caribbean Sea estimated using in situ and remote sensing data., Remote Sensing of Environment, 89, 309–325, 2004.



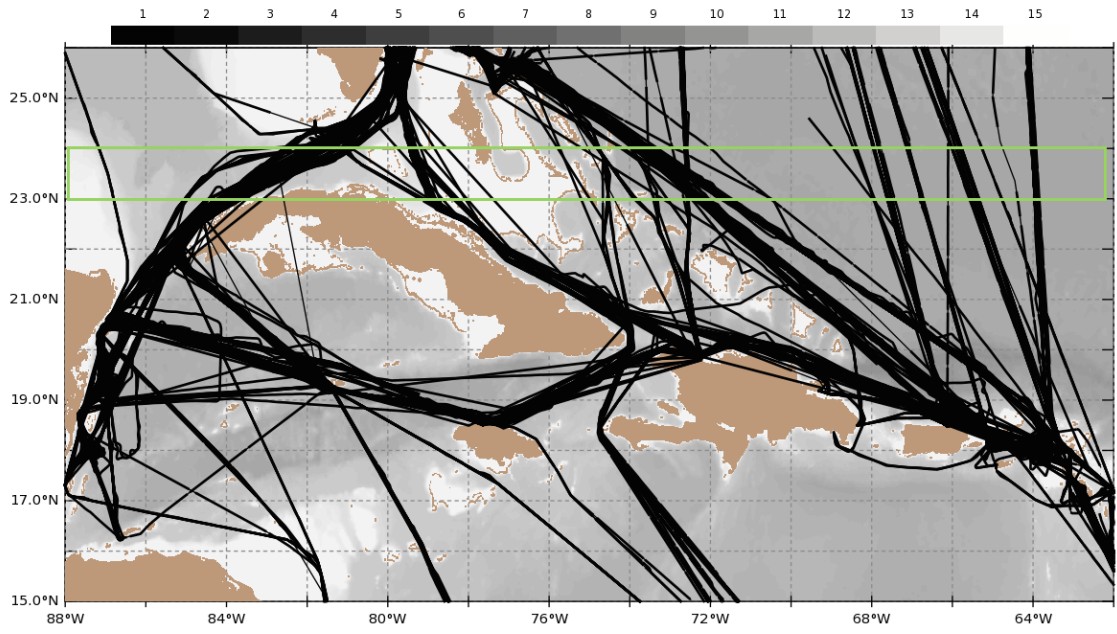


Fig 1.






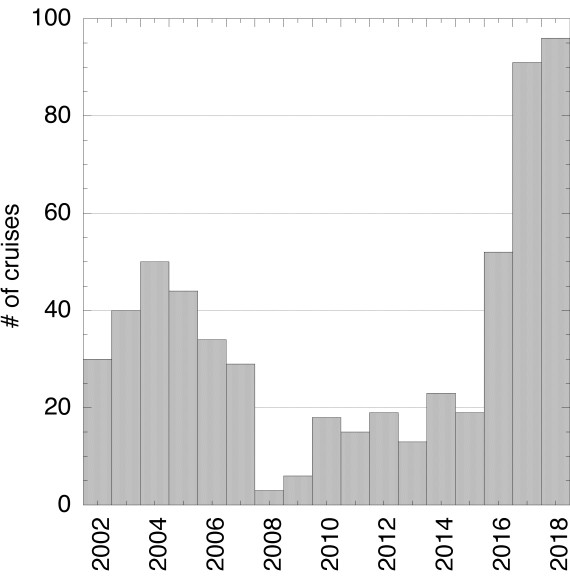

Fig 2.





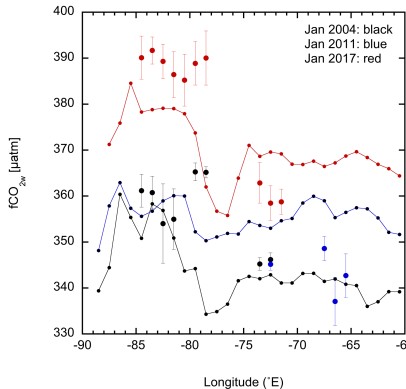

Fig 3a

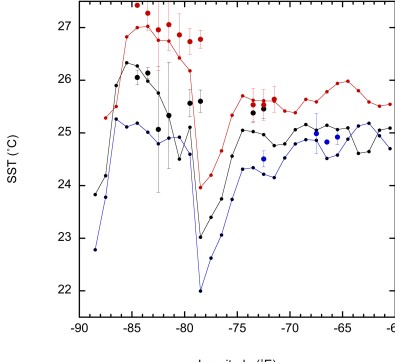

Fig 3b

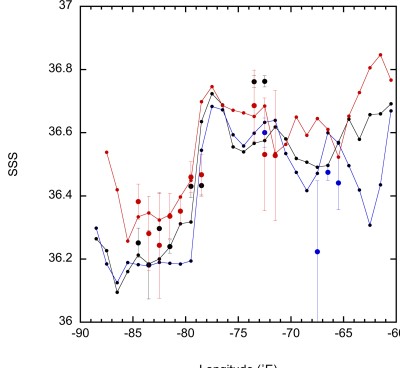

Fig 3c






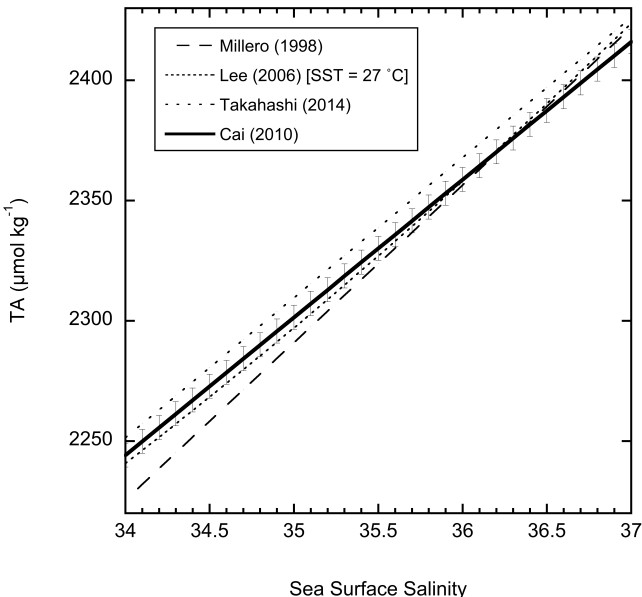

Fig 4






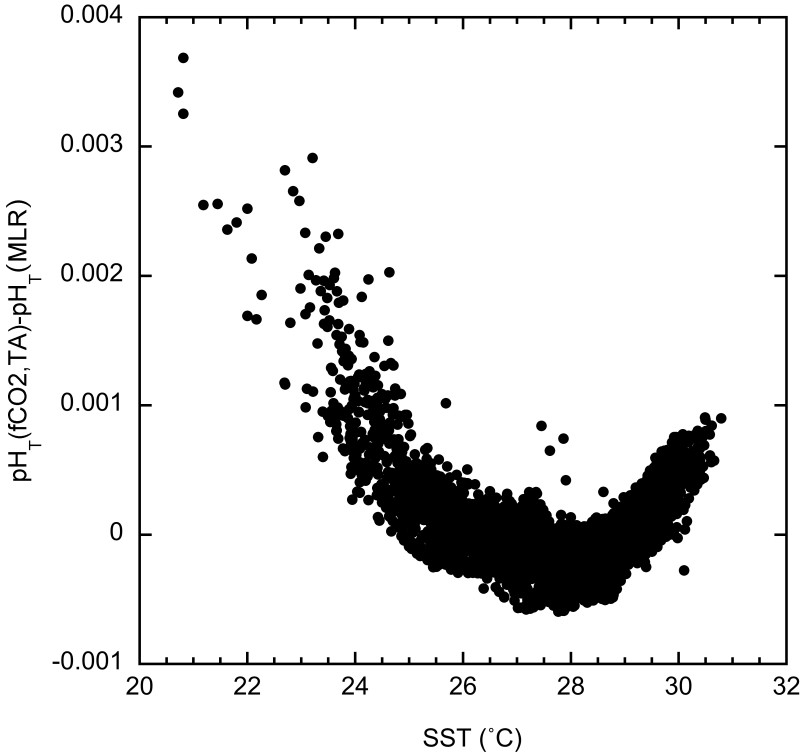

Fig 5
