# Peer review of "A 17-year dataset of surface water fugacity of CO2, along with calculated pH, aragonite saturation state, and air-sea CO2 fluxes in the Northern Caribbean Sea"

_Earth System Science Data, 2019_

## Referee Comment (RC1) · Anonymous Referee #1 · 28 Feb 2020

A 17-year dataset of surface water fugacity of CO2, along with calculated pH, Aragonite saturation state, and air-sea CO2 fluxes in the Northern Caribbean Sea.

This manuscript presents a straightforward assessment of an impressive dataset taken in the Caribbean Sea aboard two cruise ships outfitted with state-of-the-art CO2 measurement equipment. The CO2 data along with modeled salinity (used to estimate TA), MLD and satellite SST were used in a year by year algorithm to estimate pH and Omega-aragonite. A statistical binning process is used to aggregate these data into 1-degree bins.

[Figure]

This work should be good to be published with minor revisions assuming the comments are addressed.

My most pressing concern centers on using separate fitting algorithms for each year. The authors need to justify this. A quick glance reveals that the differences in the coefficients are quite large year by year. Why wouldn't it make more sense to do this by aggregating seasonal data? When considering the time series data: what would be the effect of the use of annual estimates rather than a single fitting? For example, I wouldn't expect a smooth transition between December and the subsequent January that uses a different algorithm.

Also, I do not see good description of how the derived parameters pH and Omega Ar were validated.

Minor concerns: Line 121. A Licor 6262 was used. While this is a fine instrument, it has not been production since 2005. I'm not worried about the $CO_2$ measurements since they do a good job with standards, but how was the $H_2O$ channel calibrated or standardized? How accurate is your $pH_2O$?

Section 2.4. What information do the MLD estimates convey? They mention what they could be used for e.g. inventories, but nothing about how they help the estimates. Please include your ideas on this.

Line 150. The term "bone dry" is not very scientific and should have been hyphenated. Plus, bones are not that dry. See, Timmins, P.A., Wall, J.C. Bone water. Calc. Tis Res. 23, 1–5 (1977). https://doi.org/10.1007/BF02012759

Line 150. Mentions "analyses typically had a humidity of 10% or less", but earlier (L. 120) mentioned that the headspace was dried >75%. I think I get the distinction, but it was confusing to me.

Line 267. Mentions that cooler near surface temperatures could lead to lower fCO2 values and that this has a "large impact on the calculated air sea fluxes". Please use a

citation or constrain the "Large impact" with some stats.

Line 273-275. Where does the steady state increase of 2.13 uatm/yr come from? Is this from the ship's atm data, or is it Mauna Loa, carbon tracker or something else. Also, over a 17-year time series, one would expect a changing rate of CO2atm. Is "steady state" referring to the linear increase over that time period? Clarify.

Line 324 For Boron, I though the Lee at al, 2009 algorithm was more commonly used these days.

Line 343. Use a reference or two for the use of Omega as a biological indicator.

Equation 9. If this is right out of CO2sys, I see no reason for an equation.

Figure 1. It's unfortunate that there's little data where the salinity variability is presumably the highest (i.e. in the Southern Caribbean where the large South American Rivers affect the region). What is the effect of this on the gridded data?

Acknowledgments: should mention the cruise line that made this possible.

————————————————————

---

## Referee Comment (RC2) · Anonymous Referee #2 · 1 Mar 2020

The manuscript presents a dataset of surface ocean fCO2, and auxiliary variables, measured in the Caribbean from 2002-2018. In addition, a data product consisting of gridded and gap-filled maps of fCO2, pH, aragonite saturation state, and air-sea CO2 fluxes is produced and presented. Both the observational dataset and the data products are of undoubtedly high quality and will very likely be very useful to the global ocean carbonate chemistry community. The manuscript is nicely presented and illustrated, and overall well-written though at times highly repetitive. This work is highly relevant for publication in ESSD and can be published after minor revisions (detailed

below).

Major comments:

Why use annual multilinear regressions? I understand from the appendix that using delta_fCO2 did not improve the results, but I'd like to also see what difference it would make to use one multilinear regression where atmospheric xCO2 (or fCO2) is included as a predictor variable. Have you analyzed whether the use of annual multilinear regressions create discontinuities between December and January? Please add a figure showing that this is negligible.

I find the entire manuscript to poorly structured which results in a lot of repetition. I suggest to restructure in order to create a nicer flow of information and thus increase readability. Some suggestions, in no particular order: - The information on lines 94-104 would be better suited in section 1.3 (instrumentation) - Information in section 1.3 (instrumentation) and sections 2.1 and 2.2 should be combined and the text screened for repetitive information (e.g., the frequency of calibration is mentioned on line 114 and again on line 122) - I'm not sure of the value of section 1.3.3 unless these data are used in the presented dataset or data products (which is unclear) - Section 2 could be a subsection under section 1 - Much of the information on lines 65-74 would be more appropriate in the methods (much of it is also repeated in the different methods sections) - The information on lines 270-294 would be better suited in section 3.5 - In section 4.1 you give much information which is suitable, and partly repeated, in section 5

Minor comments:

Line 64: I'd prefer the term "raw data processing" over "data reduction". While the former is commonly used in the community, it is not intuitive to those outside what it actually involves In the introduction it is stated that the Explorer of the Seas changed her home port to Bayonne, NJ in 2008 while in section 1 it is stated that the new home port is Cape Liberty Cruise Port. I realize these may be in the same place but it is nevertheless confusing. Please revise Line 171: Explain what flag questionable is (presumably WOCE 3) Line 242: I do not understand the method. Please explain. Line 340-341: While this is correct I find it helpful to instead state that when omega_Ar<1 dissolution is thermodynamically favored, and vice versa when omega_Ar>1. In living organisms both dissolution and precipitation of calcium carbonate is biologically mediated, and shells have been shown to survive well in water with omega_Ar ~0.9. In section 4 you should define the difference between a dataset and data products. My experience is that surprisingly many do not know the difference. It is unclear whether you consider the gridded data part of the dataset or a data product.
* * *

---

## Referee Comment (RC3) · Anonymous Referee #3 · 2 Mar 2020

The manuscript describes a 17 year dataset for surface water marine carbonate data collected using multiple ships within the Caribbean and a substantial set of derived data.

The manuscript appears to have been a little rushed. There are instances of unclear statements, inconsistent naming, repetition, use of non-SI units, formatting errors and some structural issues. I have listed all coments referring to these issues under the section 'Minor comments' (See below).

[Figure]

I would suggest that the manuscript is re-considered after revision and my reasoning is explained below within the Major comments.

Major comments: 1. The uncertainty information within the manuscript is inconsistent and/or incomplete. Some information is given for the fCO2 data but nothing is given for the temperature or salinity. No uncertainty information or statements are given for the derived datasets eg pH or aragonite saturation state or gas fluxes. This limited information will limit the use of these data, or could result in users making incorrect assumptions about the uncertainties. It would be good if the authors could follow a standard framework or phrasing for presenting the uncertainty information e.g. BIPM 2008 framework and identifying if uncertainties are Type A or Type B and also identify which components of the uncertainty budget have been considered and which have been ignored. Its clear that the derived datasets are unlikely to be considered to be 'truth' measurements, so the authors need to write some text to explain this, so that users of the dataset don't make the mistake of assuming that these data are truth. It may make sense, and/or make it easier for the reader, if all of the uncertainty information was grouped together into a common location (eg one table?) which can then be referred to within the different sections of the manuscript.

BIPM, 2008 - Guide to the expression of uncertainty https://www.bipm.org/utils/common/documents/jcgm/JCGM_100_2008_E.pdf

2. lines 266 to 268. The text states that the cooler temperatures near the surface could lead to lower fCO2 which can have large impact on the calculated air-sea gas fluxes. But the gas fluxes have been calculated using a version of the bulk flux calculation (equation 4) which ignores all vertical temperature gradients. However the dataset includes OISST data which could be used to perform a more accurate gas flux calculation (e.g. re-calculate pCO2 to a common depth, then perform a more accurate calculation). The authors could either provide the results using a more accurate gas flux calculation or highlight this issue to the user/reader and then refer to them to an example analysis that shows the impact of a lower accuracy gas flux calculation and

the estimate the increased uncertainty within their derived dataset that results from this lower accuracy calculation. To help, see figures 3 and figure 4 of Holding et al., (2019) for an analysis of the impact along single cruise tracks, or panel 1 of Shutler et al., (2019) for the impact over larger spatial and temporal areas.

Holding et al., (2019) https://www.ocean-sci.net/15/1707/2019/ Shutler et al. (2019) https://esajournals.onlinelibrary.wiley.com/doi/pdf/10.1002/fee.2129

3. its not clear why the multi-linear regressions are performed and/or why anyone would want this output. These results and methods should introduced giving an explanation as to why they are useful. I'm not sure that this part of the dataset is needed though.

4. The binning method does not account for the paired nature of the pCO2 and SST datasets (as each parameter is binned individually). Surely the binning will have skewed this relationship and so the paired nature will no longer exist. This issue may be especially true if some bins contain data from multiple cruises (which fig 1 suggests will occur). Can the authors highlight this issue and discuss the implications so that users of the dataset are aware of this problem?

Minor comments: 1. line 30, suggest 'The data and products could be used for determination of ....' as surely the paper is providing data for others to use (rather than presenting their use of these data). 2. the use of the word 'average' throughout the manuscript is ambiguous. do the authors mean a statistical mean, mode or median? (all are averages). suggest that all instances of the word 'average' are replaced with the appropriate statistical name. 3. there are instances of 'month' and 'mo'. the latter I think also means 'month'. I'd suggest that the authors use one throughout, rather than swapping between both. 4. line 99, space needed between 100 and units (m). 5. line 101, no dash needed in '5-m'. similarly three further instances on line 109 and more instances of this on line 191. 6. line 126, the value of 2uatm is twice the size of the value on line 120. are these the same values ie +-1uatm? how has this value of 2uatm been estimated? 7. line 115, I'd suggest '...measurements for the ships with

air intakes and analysers.' 8. line 124. can you provide the range in values used for the standards? 9. section 1.3.2 the precision, accuracy and sensitivity of these instruments are missing. 10. line 150, use of non scientific phrasing, what is 'bone dry'? 11. line 158, can you define what you mean by infrequently? eg. %age of time. 12. line 160. processing routines are mentioned but no detail is given. could an overview of these processing routines be provided in the appendices? this information would appear fairly important should anyone want to use these data and/or try and follow the same methods for a similar effort somewhere else in the world. 13. line 266, I'd suggest 'While both differences include zero within their uncertainty.' 14. Suggest that section 3.4 (binning procedure) comes before the sections on the calculations (as surely the binning is done first, then the calculations are performed). 15. line 303, I think that CCMP data are available for 2018 eg http://data.remss.com/ccmp/v02.0/ 16. line 327, see 'were compared for 201'. what is 201? 17. line 336, 'insignificant' is a bit subjective and application specific. can you put this into context? 18. table 3, space needed between 12 and (December). 19. table 3, 4, and 5 all contain non-SI unit notation in pH row (mol/kg-SW) 20. line 407, the 'stdev' has previously been used, but not defined. 21. the content in section 4.5 is a bit jumbled. The method for the annual and monthly values needs to be more clearly and sequentially explained. Eg surely the values are first weighted by area and then summed (rather than summed and the mean value area weighted?). 22. line 411 to 414. Can you clarify this paragraph? I'm afraid that I don't really understand this paragraph or the reasoning and why are the data treated differently? 23. line 409, the dash between terra and grams is not needed. 24. section 5 contains repetition (with section 4.1). 25. line 445, I'd suggest '..instrumental in maintaining the science operations.' 26. line 667, month/year notation is different from the main paper. 27. line 676, how is this 'overall uncertainty' determined? 28. line 677, mixing of 'errors' and 'uncertainties' naming. I think that they are all uncertainties (error implies that you know a truth value). What is the 'error' column in table 1? and it appears to be called RMSE in table A1 and A2. 29. line 683, Lat and Lon not defined. 30. line 684, RMS not defined.

---

## Author Comment (AC1) · 27 Apr 2020

Note, As formatting was lost in the plain text, the response is also attached as a pdf

We thank the reviewers for the time taken to carefully read the manuscript and the thorough review and comments. We have incorporated most in the edited copy and we feel that these changes have markedly improved the manuscript. We apologize for the somewhat sloppy and sometimes repetitive nature of the manuscript. The paper was created from separate contributions of the co-authors, and it also "suffered" from a late

decision to provide this dedicated manuscript on the data. Parts were cut from the draft of the manuscript describing the science. (Wanninkhof, R., Trinanes, J. A., Park, G.-H., Gledhill, D. K., and Olsen, A.: Large Decadal Changes in Air-Sea CO2 Fluxes in the Caribbean Sea„ Journal of Geophysical Research, 10.1029/2019JC015366, 2019.). We have eliminating the repetitive sections. While all the review comments were very useful, some are outside the scope of ESSD on the presentation of the datasets (see below). For the reviewers comments that pertain to interpretation we largely refer to the 2019 paper, or reference other key papers. We have referred to the Wanninkhof et al. 2019 paper several more times in the edits when appropriate. Specific responses are listed below embedded in the reviewers comments.

"Scope of ESSD: "Earth System Science Data (ESSD) is an international, interdisciplinary journal for the publication of articles on original research data (sets), furthering the reuse of high-quality data of benefit to Earth system sciences. The editors encourage submissions on original data or data collections which are of sufficient quality and have the potential to contribute to these aims."
A 17-year dataset of surface water fugacity of CO2, along with calculated pH, Aragonite saturation state, and air-sea CO2 fluxes in the Northern Caribbean Sea. This manuscript presents a straightforward assessment of an impressive dataset taken in the Caribbean Sea aboard two cruise ships outfitted with state-of-the-art CO2 measurement equipment. The CO2 data along with modeled salinity (used to estimate TA), MLD and satellite SST were used in a year by year algorithm to estimate pH and Omega-aragonite. A statistical binning process is used to aggregate these data into 1-degree bins.

This work should be good to be published with minor revisions assuming the comments are addressed. My most pressing concern centers on using separate fitting algorithms for each year. The authors need to justify this. A quick glance reveals that the differences in the coefficients are quite large year by year. Why wouldn't it make more sense to do this by aggregating seasonal data? When considering the time series data: what would be the effect of the use of annual estimates rather than a single fitting? For example, I wouldn't expect a smooth transition between December and the subsequent January that uses a different algorithm. We have provided additional justification and show that there are no large offsets from year to year fit in a new figure: "The MLRs to create the monthly mapped products were produced for each year such that the mapped products could be extended each year in a straightforward fashion. To determine if there were anomalous discontinuities between December and January that could impact the timeseries, the difference in between fCO2w for subsequent months were plotted versus time in Figure 3. No significant discontinuities were observed. Only for Jan 2009, 2010, and 2017 there appear to be a slight difference in the pattern of monthly progressions but such anomalies are observed during other times of year as well. Using an MLR that includes year as one of the coefficients (supplemental material S2 and S3) provides a slightly worse fit. More importantly, using such a fit would necessitate recalculating the mapped products every time a new year is added. "

Also, I do not see good description of how the derived parameters pH and Omega Ar were validated. Since Omega is calculated, and pH was not measured on the cruise there is no straightforward way to validate these products. We have emphasized that these are products and have added a section and a new Table 2 on estimated uncertainties of the products.

Minor concerns: Line 121. A Licor 6262 was used. While this is a fine instrument, it has not been production since 2005. I'm not worried about the CO2 measurements since they do a good job with standards, but how was the H2O channel calibrated or standardized? How accurate is your pH20? We have listed that the H2O channel

was zeroed, and that water vapor concentration was very low Section 2.4. What information do the MLD estimates convey? They mention what they could be used for e.g. inventories, but nothing about how they help the estimates. Please include your ideas on this. We have detailed that this is a gap filling technique Line 150. The term "bone dry" is not very scientific and should have been hyphenated. Plus, bones are not that dry. See, Timmins, P.A., Wall, J.C. Bone water. Calc. Tis Res. 23, 1–5 (1977). https://doi.org/10.1007/BF02012759. This is a common expression in the "old" chemistry school. We have eliminated this phrase Line 150. Mentions "analyses typically had a humidity of 10% or less", but earlier (L. 120) mentioned that the headspace was dried >75%. I think I get the distinction, but it was confusing to me. This was corrected. It is less than 10 % Line 267. Mentions that cooler near surface temperatures could lead to lower fCO2 values and that this has a "large impact on the calculated air sea fluxes". Please use a citation or constrain the "Large impact" with some stats. We added the estimate from the Wanninkhof 2019 paper. "As shown in Wanninkhof et al. 2019b, a 0.25 ËŽC bias in SST leads to a 2 $\mu$atm difference in fCO2w and a 25 % change in the air-sea CO2 flux" and we also provide a brief discussion in the new section on uncertainties, including two new references

Line 273-275. Where does the steady state increase of 2.13 uatm/yr come from? Is this from the ship's atm data, or is it Mauna Loa, carbon tracker or something else. Also, over a 17-year time series, one would expect a changing rate of CO2atm. Is "steady state" referring to the linear increase over that time period? Clarify. The source of atmospheric data was provided two paragraphs below line 273-275. "For the air-sea CO2 fluxes, the monthly fCO2a values were derived from the weekly average xCO2a of the stations on Key Biscayne (KEY) and Ragged Point Barbados (RPB) (Carbon-Tracker Team, 2019; https://www.esrl.noaa.gov/gmd/ccgg/flask.php). We changed the sentence to" At steady state this atmospheric CO2 increase would translate to a linear trend of fCO2w of 2.13 $\mu$atm yr-1 over the time period."

Line 324 For Boron, I though the Lee at al, 2009 algorithm was more commonly used

these days. The verdict is still out. In the SOCCOM project and in Orr et al. 2018 (new reference) the use of Uppström (1974) is recommended. We left this unchanged. Line 343. Use a reference or two for the use of Omega as a biological indicator. We added one specific to corals: Mollica, N. R., Guo, W., Cohen, A. L., Huang, K.-F., Foster, G. L., Donald, H. K., and Solow, A. R.: Ocean acidification affects coral growth by reducing skeletal density, Proceedings of the National Academy of Sciences, 115, 1754, 10.1073/pnas.1712806115, 2018. Equation 9. If this is right out of CO2sys, I see no reason for an equation. You would have to dig into the code of CO2Sys and this equation is pretty fundamental, but seldom presented, when discussing saturation states. We left this unchanged.

Figure 1. It's unfortunate that there's little data where the salinity variability is presumably the highest (i.e. in the Southern Caribbean where the large South American Rivers affect the region). What is the effect of this on the gridded data? As shown in Wanninkhof 2019 (figures 3 and 4) and accompanying text, the salinity anomalies show up in the SE and NW part of the region. However, these variabilities in salinity have a small effect on the total area. Acknowledgments: should mention the cruise line that made this possible. Yes, thanks for pointing this out. We had RCCL listed in the team list but we have added the following in the acknowledgement : "This work would not have been possible without support from Royal Caribbean Cruise Lines who have provided access to their ships and significant financial, personnel, and infrastructure resources for the measurement campaign coordinated through the Rosenstiel School of Marine and Atmospheric Sciences of the University of Miami."
The manuscript presents a dataset of surface ocean fCO2, and auxiliary variables, measured in the Caribbean from 2002-2018. In addition, a data product consisting of

gridded and gap-filled maps of fCO2, pH, aragonite saturation state, and air-sea CO2 fluxes is produced and presented. Both the observational dataset and the data products are of undoubtedly high quality and will very likely be very useful to the global ocean carbonate chemistry community. The manuscript is nicely presented and illustrated, and overall well-written though at times highly repetitive. This work is highly relevant for publication in ESSD and can be published after minor revisions (detailed below)

Major comments: Why use annual multilinear regressions? I understand from the appendix that using delta_fCO2 did not improve the results, but I'd like to also see what difference it would make to use one multilinear regression where atmospheric xCO2 (or fCO2) is included as a predictor variable. Have you analyzed whether the use of annual multilinear regressions create discontinuities between December and January? Please add a figure showing that this is negligible.

This was noted by reviewer 1 as well. We have provided additional justification and show that there are no large offsets between year to year fits: "The MLRs to create the monthly mapped products were produced for each year such that the mapped products could be extended each year in a straightforward fashion. To determine if there were anomalous discontinuities between December and January that could impact the time-series, the difference in between fCO2w for subsequent months were plotted versus time in Figure 3. No significant discontinuities were observed. Only for Jan 2009, 2010, and 2017 there appear to be a slight difference in the pattern of monthly progressions but such anomalies are observed during other times of year as well. Using an MLR that includes year as one of the coefficients (supplemental material) provides a worse fit. More importantly, using such a fit would necessitate recalculating the mapped products every time a new year is added. "

I find the entire manuscript to poorly structured which results in a lot of repetition. I suggest to restructure in order to create a nicer flow of information and thus increase readability. Some suggestions, in no particular order: - The information on lines 94-

104 would be better suited in section 1.3 (instrumentation) - Information in section 1.3 (instrumentation) and sections 2.1 and 2.2 should be combined and the text screened for repetitive information (e.g., the frequency of calibration is mentioned on line 114 and again on line 122) While we have retained the structure, but we have rearranged the text and eliminated repetition as shown the text using MS-Word track changes and it is more readable now . - I'm not sure of the value of section 1.3.3 unless these data are used in the presented dataset or data products (which is unclear) This is to indicate that this UWpCO2 system is part of a larger effort. This also points readers to a possible opportunity to utilize these observations in conjunction with UWpCO2 data and data products presented

- Section 2 could be a subsection under section 1 - Much of the information on lines 65-74 would be more appropriate in the methods (much of it is also repeated in the different methods sections) - The information on lines 270-294 would be better suited in section 3.5 - In section 4.1 you give much information which is suitable, and partly repeated, in section 5 We've made significant edits along the lines suggested but retained the overall structure. E.g. section 1. Is about instrumentation, section 2 is about data from the instrumentation. For readers who are more interested in the data it avoids needing to closely read the detail of the instruments themselves. Minor comments: Line 64: I'd prefer the term "raw data processing" over "data reduction". While the former is commonly used in the community, it is not intuitive to those outside what it actually involves We are confused by this comment as it suggests that the recommended change to "raw data processing" is not intuitive. We agree with that as "raw data" is a bit ambiguous and we kept nomenclature as is. In the introduction it is stated that the Explorer of the Seas changed her home port to Bayonne, NJ in 2008 while in section 1 it is stated that the new home port is Cape Liberty Cruise Port. I realize these may be in the same place but it is nevertheless confusing. Thanks for pointing this out it was changed

Please revise Line 171: Explain what flag questionable is (presumably WOCE 3) Done

Line 242: I do not understand the method. Please explain. Line 340-341: While this is correct I find it helpful to instead state that when omega_Ar<1 dissolution is thermodynamically favored, and vice versa when omega_Ar>1. In living organisms both dissolution and precipitation of calcium carbonate is biologically mediated, and shells have been shown to survive well in water with omega_Ar 0.9. Based on this reviewers comments, and that of reviewer one, we have changed the text to "That is, when ΩAr is less than 1 dissolution is thermodynamically favored and when ΩAr is greater than 1 Aragonite would have a tendency to precipitate."

In section 4 you should define the difference between a dataset and data products. My experience is that surprisingly many do not know the difference. It is unclear whether you consider the gridded data part of the dataset or a data product. We've adopted following nomenclature [following Wanninkhof et al. 2019 ]and checked for consistency thorough the paper: Data : individual data points (observations) Gridded data: binned and averaged data ( in this case monthly on a 1 by 1 grid) Gridded data product: a derived (calculated or interpolated) ( in this case monthly on a 1 by 1 grid) Mapped data (product): interpolated using the MLRs
The manuscript describes a 17 year dataset for surface water marine carbonate data collected using multiple ships within the Caribbean and a substantial set of derived data. The manuscript appears to have been a little rushed. There are instances of unclear statements, inconsistent naming, repetition, use of non-SI units, formatting errors and some structural issues. I have listed all comments referring to these issues under the section 'Minor comments' (See below). These issues were also brought up by the other reviewers and we have addressed these and/or provided detail about changes (or why not) below. The issue of non-SI units is noted but we use the community accepted terms of expressing fCO2 in $\mu$atm and pressure in millibar. This is done almost exclusively in ocean carbon cycle research.

I would suggest that the manuscript is re-considered after revision and my reasoning is explained below within the Major comments. Major comments: 1. The uncertainty information within the manuscript is inconsistent and/or incomplete. Some information is given for the fCO2 data but nothing is given for the temperature or salinity. No uncertainty information or statements are given for the derived datasets eg pH or aragonite saturation state or gas fluxes. This limited information will limit the use of these data, or could result in users making incorrect assumptions about the uncertainties. It would be good if the authors could follow a standard framework or phrasing for presenting the uncertainty information e.g. BIPM 2008 framework and identifying if uncertainties are Type A or Type B and also identify which components of the uncertainty budget have been considered and which have been ignored. Its clear that the derived datasets are unlikely to be considered to be 'truth' measurements, so the authors need to write some text to explain this, so that users of the dataset don't make the mistake of assuming that these data are truth. It may make sense, and/or make it easier for the reader, if all of the uncertainty information was grouped together into a common location (eg one table?) which can then be referred to within the different sections of the manuscript. BIPM, 2008 - Guide to the expression of uncertainty https://www.bipm.org/utils/common/documents/jcgm/JCGM_100_2008_E.pdf

We appreciate the recognition of the reviewer of the importance of characterizing uncertainty that is lacking in many manuscripts covering the environmental sciences, including this one. The guide is also appreciated and read (during this time of shelter at home). We have added a section of uncertainty estimates on the data products following the nomenclature and approach outlined in Orr et al. 2018.. The uncertainties in the observations using the instrumentation described and the calculated fCO2w has been described in Pierrot et al, 2006. While full characterization of the uncertainty in the data products is challenging we feel that this new section is an important addition.

lines 266 to 268. The text states that the cooler temperatures near the surface could lead to lower fCO2 which can have large impact on the calculated air-sea gas fluxes. But the gas fluxes have been calculated using a version of the bulk flux calculation (equation 4) which ignores all vertical temperature gradients. However the dataset includes OISST data which could be used to perform a more accurate gas flux calculation (e.g. re-calculate pCO2 to a common depth, then perform a more accurate calculation). The authors could either provide the results using a more accurate gas flux calculation or highlight this issue to the user/reader and then refer to them to an example analysis that shows the impact of a lower accuracy gas flux calculation and he estimate the increased uncertainty within their derived dataset that results from this lower accuracy calculation. To help, see figures 3 and figure 4 of Holding et al., (2019) for an analysis of the impact along single cruise tracks, or panel 1 of Shutler et al., (2019) for the impact over larger spatial and temporal areas. Holding et al., (2019) https://www.ocean-sci.net/15/1707/2019/ Shutler et al. (2019) https://esajournals.onlinelibrary.wiley.com/doi/pdf/10.1002/fee.2129

This is a description of the mapped product using the conventional bulk flux parameterization as described. This is the same approach as in most climatologies. We are aware of the developments and claims of superior ways of determining the fluxes by normalizing to a common depth and using skin temperatures. We have added a short description in the uncertainty section and referenced two key papers addressing the cool skin effect controversy. This is mentioned in the Wanninkhof 2019 paper as well .

its not clear why the multi-linear regressions are performed and/or why anyone would want this output. These results and methods should introduced giving an explanation as to why they are useful. I'm not sure that this part of the dataset is needed though. We have clarified that this is the means of mapping for gap filling. As mentioned:" The gridded observations (1ËŽ by 1ËŽ by mo) represent about 10 % of the area of investigation from 15-28 ËŽN and 88-62 ËŽW over the period of investigation" so there has to be a means to fill in the missing cells. As described in Wanninkhof et al. 2019,

and here the variation in each grid cell provided: "The standard deviation (stdev) of the fCO2w in each cell is determined and then the average of the stdev for the 9924 cells with observations is taken. The average stdev is 3.4 ± 2.6 $\mu$atm (n=9224) indicating the small variability in each cell. The same procedure is followed for SST and SSS yielding values of 0.22± 0.19 ËŽC for SST; and 0.10± 0.10 for SSS. These are relatively small deviations compared to the monthly spatial range of ≈ 20 $\mu$atm for fCO2w; ≈1 ËŽC for SST; ≈ 1 in SSS. The amplitude of the seasonal cycle of ≈ 40 $\mu$atm for fCO2w and ≈4 ËŽC for SST is significantly greater than the average stdev as well 2. The binning method does not account for the paired nature of the pCO2 and SST datasets (as each parameter is binned individually). Surely the binning will have skewed this relationship and so the paired nature will no longer exist. This issue may be especially true if some bins contain data from multiple cruises (which fig 1 suggests will occur). Can the authors highlight this issue and discuss the implications so that users of the dataset are aware of this problem? We are not completely clear what the reviewer is getting at. As listed the stdev of the binned and averaged data is quite small and the analysis does not rely on the paired nature of pCO2 and SST. "The standard deviation (stdev) of the fCO2w in each cell is determined and then the average of the stdev for the 9924 cells with observations is taken. The average stdev is 3.4 ± 2.6 $\mu$atm (n=9224) indicating the small variability in each cell. The same procedure is followed for SST and SSS yielding values of 0.22± 0.19 ËŽC for SST; and 0.10± 0.10 for SSS. These are relatively small deviations compared to the monthly spatial range of ≈ 20 $\mu$atm for fCO2w; ≈1 ËŽC for SST; ≈ 1 in SSS. The amplitude of the seasonal cycle of ≈ 40 $\mu$atm for fCO2w and ≈4 ËŽC for SST is significantly greater than the average stdev as well" Minor comments: 1. line 30, suggest 'The data and products could be used for de- termination of ....' as surely the paper is providing data for others to use (rather than presenting their use of these data). This merely provides examples of possible use.

2. the use of the word 'average' throughout the manuscript is ambiguous. do the authors mean a statistical mean, mode or median? (all are averages). suggest that

all instances of the word 'average' are replaced with the appropriate statistical name. As listed in section 4.2.1 of the BIPM, 2008 - Guide to the expression of uncertainty. "arithmic mean or average". When we refer to average are referring to the arithmic mean. I am not familiar with calling mode or median an average

3. there are instances of 'month' and 'mo'. the latter I think also means 'month'. I'd suggest that the authors use one throughout, rather than swapping between both. 4.

Thanks. We only use "mo" in units and when describing the grid cell and this is mentioned the first time it is used : "For the regionally mapped products on a 1-degree grid and monthly timescale (1ËŽ by 1ËŽ by mo)". The grid size (1ËŽ by 1ËŽ by mo) is listed a lot in part based on a reviewers request to the JGR Wanninkhof et al. 2019 paper. For clarity we have placed brackets around each occurrence of (1ËŽ by 1ËŽ by mo)".

line 99, space needed between 100 and units (m). Thanks, done 3. line 101, no dash needed in '5-m'. similarly three further instances on line 109 and more instances of this on line 191. Thanks, corrected I am always confused when a dash is needed

4. line 126, the value of 2uatm is twice the size of the value on line 120. are these the same values ie +-1uatm? how has this value of 2uatm been estimated? This is provided in the reference that is now added , "estimated at better than 2 $\mu$atm (Pierrot et al., 2009)," The intercomparison showing 1 uatm agreement used common standards, pressure and temperature measurements. The 2 uatm incorporates uncertainties in Teq and P as described in the provided reference.

5. line 115, I'd suggest '...measurements for the ships with ir intakes and analysers.' 8. line 124. can you provide the range in values used for the standards? Added in the description of the standards " The four calibration gases supplied by the global monitoring division of the environmental science research laboratory of NOAA (GMD/ ESRL/NOAA) are traceable to the WMO CO2 mole fraction scale. The CO2 concentrations of the standards span the range of surface water values encountered along the ship tracks ($\approx$ 280- 480 ppm),.

6. section 1.3.2 the precision, accuracy and sensitivity of these instruments are missing. We have now stated that these can be found in Pierrot et al (2009). However, we do not include sensitivity

7. line 150, use of non scientific phrasing, what is 'bone dry'? This has been replaced . As noted, "bone dry" used to be the common terminology: "bone-dry=completely dry"

8. line 158, can you define what you mean by infrequently? eg. %age of time. Change to "Infrequently (<1 % of the time)"

9. line 160. processing routines are mentioned but no detail is given. could an overview of these processing routines be provided in the appendices? this information would appear fairly important should anyone want to use these data and/or try and follow the same methods for a similar effort somewhere else in the world. The processing routines are described in Pierrot et al. (2006) and as listed MATLAB routines are available from Pierrot on request. 10. line 266, I'd suggest 'While both differences include zero within their uncertainty..' Done 11. Suggest that section 3.4 (binning procedure) comes before the sections on the calculations (as surely the binning is done first, then the calculations are performed). We've rearranged the sections and clarified procedures. The fCO2 and pCO2 was calculated and then the results were binned. For the fluxes we used the binned product. So placing it before the fluxes would make sense but it does not flow as well. Added a reference to the following section "a bulk formulation is applied to the data from the gridded mapped product (see 3.4):"

12.     line 303, I think that CCMP data are available for 2018 eg http://data.remss.com/ccmp/v02.0/ Yes but this was not available when the calculations were performed. These will be annual updates and when the 2019 data is included the winds will be updated 13. line 327, see 'were compared for 201'. what is 201? Corrected to 2017 14. line 336, 'insignificant' is a bit subjective and application specific. can you put this into context? We have changed this and included it in the discussion of uncertainty "This MLR was then applied to the independent variables for

each grid box to determine pHT(MLR). This was compared to the approach used here of calculating the pH using the mapped fCO2wMLR and TA-SSS relationships on (1ËŽ x 1ËŽ x mo) grids, called pHT(fCO2w,TA). The two approaches provided similar results with pHT(fCO2w,TA) - pHT(MLR) = -0.0001 ± 0.005 for 2017. The small difference showed a pattern with SST (Fig. 6) but not with the other independent variables. The differences using either pHT(fCO2w,TA) or pHT,MLR are an order of magnitude smaller than the combined standard uncertainties such that the approach of using mapped fCO2w and TA to determine pHT(MLR) yielded precise and consistent gridded pHT." 15. table 3, space needed between 12 and (December). done 16. table 3, 4, and 5 all contain non-SI unit notation in pH row (mol/kg-SW) unclear pH is unitless as indicated 17. line 407, the 'stdev' has previously been used, but not defined. Defined in section 3.4: " The standard deviation (stdev) of" 18. the content in section 4.5 is a bit jumbled. The method for the annual and monthly values needs to be more clearly and sequentially explained. Eg surely the values are first weighted by area and then summed (rather than summed and the mean value area weighted?). Thanks. This section has been rewritten 19. line 411 to 414. Can you clarify this paragraph? I'm afraid that I don't really understand this paragraph or the reasoning and why are the data treated differently? Thanks. This section has been rewritten 20. line 409, the dash between terra and grams is not needed. Done 21. section 5 contains repetition (with section 4.1). The data availability section is a requirement for ESSD and it also must appear in the main text. 22. line 445, I'd suggest '..instrumental in maintaining the science operations.' Done 23. line 667, month/year notation is different from the main paper. We've made it consistent 24. line 676, how is this 'overall uncertainty' determined? Changed to combined Standard uncertainty in the fCO2w,MLR as per Orr et al, 2018 25. line 677, mixing of 'errors' and 'uncertainties' naming. I think that they are all uncertainties (error implies that you know a truth value). What is the 'error' column in table 1? and it appears to be called RMSE in table A1 and A2. We have defined the abbreviations in the footnote. Error is standard error 26. line 683, Lat and Lon not defined. Corrected and changed LAT and LON in tables to Lat and Lon 27.

line 684, RMS not defined. Corrected

Please also note the supplement to this comment:
https://www.earth-syst-sci-data-discuss.net/essd-2019-245/essd-2019-245-AC1-supplement.pdf

---

## Author Response (AR2)

**Topical Editor Decision: Publish subject to minor revisions (review by editor)** (15 May 2020)
by David Carlson
Comments to the Author:
Very good response and changes / upgrades.

Small remaining issues (including a few typographical changes that proofreading will probably NOT pick up).

1) Wherever you use the DOI to point to these data, please use this form instead: https://www.doi.org/10.25921/2swk-9w56. Most journals now adopt this slightly-revised format because - for most users - the new format allows one-click access whereas the prior format required cut and paste or search through DataCite.
**Response: thanks, we have changed "doi:" to https://www.doi.org/ throughout text and references**

2) Thank you for mention of FAIR principles (e.g. line 63). But, FAIR deals primarily with access, while this paper - to its credit (!) - deals almost entirely with quality. Perhaps, to follow the mention of FAIR, add some short reference to your own substantial work on quality? E.g. FAIR principles supplemented by substantial quality control efforts by this/our group. We do not want to allow an impression that meeting FAIR principles somehow substitutes for or replaces expert efforts on any data, but especially not on ocean CO2 data.
**Response: We appreciate the editor's recognition of the substantial quality controls and checks that were performed on the data. We've added the following sentence**
**"In addition, the data underwent substantial quality control by our group and through the SOCAT quality control and check procedures**."

Line 74 - calculate air-sea CO2 fluxes, rather than calculated (correct in the second use in that sentence)?
**Response: Corrected**
Line. 113 - instruments 'have performed'. Do the authors mean performed under these conditions for these data, or have performed to specs in other oceanographic applications?
**Response we have changed the sentence to :" and for the cruises described here, they have performed to high accuracy specifications at described in Wanninkhof (2013)."**
Line 418 - "For Flux calculations" should be 'For flux calculations'?
**Response: Corrected**

**In addition we corrected a few typos and punctuation errors throughout.**